# REAR: Test-time Preference
# Realignment through Reward Decomposition

**Fuxiang Zhang** [1]  **Pengcheng Wang** [2]  **Chenran Li** [2]  **Yi-Chen Li** [3]  **Yuxin Chen** [2]  **Lang Feng** [1]  **Chenfeng Xu** [2]
**Masayoshi Tomizuka** [2]  **Bo An** [1]

## Abstract

Aligning large language models (LLMs) with diverse user preferences is a critical yet challenging task. While post-training methods can adapt models to specific needs, they often require costly data curation and additional training. Test-time scaling (TTS) presents an efficient, training-free alternative, but its application has been largely limited to verifiable domains like mathematics and coding, where response correctness is easily judged. To extend TTS to preference alignment, we introduce a novel framework that models the task as a realignment problem, since the base model often fails to sufficiently align with the stated preference. Our key insight is to decompose the underlying reward function into two components: one related to the question and the other to preference information. This allows us to derive a REAlignment Reward (REAR) that selectively rescales the proportions of these two reward terms. We then show that REAR can be formulated as a linear combination of token-level policy log-probabilities, making it computationally efficient and easy to integrate with various TTS algorithms such as best-of-$N$ sampling and tree search. Experiments show that compared to other test-time baselines, REAR not only enables scalable test-time realignment for preference alignment tasks under diverse user requirements, but also generalizes to mathematical and visual tasks under appropriate preference settings.

## 1. Introduction

The remarkable success of Large Language Models (LLMs) in aligning with human preferences is largely attributed to techniques such as Reinforcement Learning from Human Feedback (RLHF) (Ouyang et al., 2022; Bai et al., 2022; Rafailov et al., 2023; Guo et al., 2025). This alignment enables a wide range of applications, from personalized assistants (OpenAI, 2023; Chen et al., 2024; Cui et al., 2024) to recommendation systems (Wu et al., 2024; Xue et al., 2023). However, a fundamental challenge remains: the preference alignment of a pretrained model is inherently tied to its training data. This often leads to a mismatch when the model is applied to downstream tasks that require personalized or diverse preferences (Jang et al., 2023; Zhang et al., 2025b;d). While this gap can be bridged through task-specific post-training (Zhang et al., 2025b; Li et al., 2025b), such methods demand significant investment in data curation and computational resources.

To circumvent the costs of post-training, we explore aligning models at inference time. While some approaches modify the policy distribution at the token level to reflect user preferences (Zhang et al., 2025c; Gao et al., 2024), they often impose a fixed computational overhead that limits their ability to scale performance with increased computational budgets. A more promising direction is Test-Time Scaling (TTS) (OpenAI, 2024; Muennighoff et al., 2025; Beeching et al., 2025), where models leverage additional computation during generation to enhance output quality. While test-time alignment has been explored through reward-guided search (Khanov et al., 2024), value guidance (Liu et al., 2024b), or speculative rejection (Sun et al., 2024a), the recent paradigm of TTS has predominantly focused on domains such as mathematics and coding, where the correctness can be easily verified (OpenAI, 2024). Applying test-time methods to preference alignment is more challenging, as the quality of a response is holistic and not reducible to a simple verifiable answer. This raises a challenge: how can we effectively guide a test-time algorithm to score model outputs and choose better responses for complex preference alignment tasks?

In this work, we address this challenge by framing the TTS

---

[1]Nanyang Technological University  [2]University of California, Berkeley  [3]Nanjing University . Correspondence to: Bo An <boan@ntu.edu.sg>.

*Proceedings of the 43rd International Conference on Machine Learning*, Seoul, South Korea. PMLR 306, 2026. Copyright 2026 by the author(s).

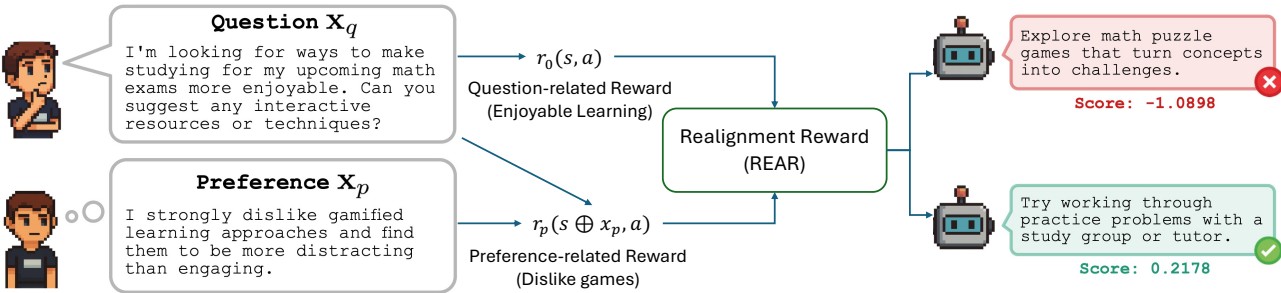

*Figure 1.* A motivating example of REAR. REAR combines the question-related reward and preference-related reward from the given text. Our method assigns REAR scores to multiple responses and finally realigns the response from a gamified suggestion that is rejected due to the user's preference to a collaborative one with a higher REAR score.

process as a realignment problem. We posit that while a pretrained model possesses general instruction-following abilities, its original training objective may not be optimal for a specific user's needs. An inference-time realignment process can rescale the importance of user preference to generate a more aligned response. As illustrated in Figure 1, when a user asks for enjoyable ways to study math but expresses a dislike for gamification, the model may sample multiple responses for this question. Some responses may only focus on answering the question of "enjoyable learning approaches", while preferred responses should also align with the preference "dislike games". Our REAlignment Reward (REAR) is designed to capture preference-alignment capabilities. Specifically, we decompose the reward of a pretrained LLM into a question-related component and a preference-related component. REAR then rescales the preference component to obtain a realigned reward value, allowing us to score candidate responses and effectively select the most aligned option. We show that REAR can be efficiently computed as a linear combination of policy log-probabilities at the token level, and thus derive a corresponding score function at the sequence level. We incorporate the REAR score into two test-time methods: a simple best-of-$N$ sampling strategy (Stiennon et al., 2020) and a more sophisticated tree search algorithm, DVTS (Beeching et al., 2025). The contributions of this paper are summarized as follows:

- We formalize test-time preference alignment as a realignment problem and propose REAR, a computationally efficient reward derived from a decomposed preference-alignment objective.

- We develop two scalable test-time algorithms guided by REAR, including best-of-$N$ sampling and DVTS, which can enhance generation performance with varying sample budgets.

- Experiments on preference-alignment and role-play

benchmarks show that our REAR-guided test-time algorithms outperform existing preference-alignment approaches. Moreover, by specifying appropriate preference text, our method generalizes to more specific mathematical and visual tasks.[1]

## 2. Preliminaries

In this section, we first formalize the text generation problem as a Markov Decision Process (MDP) (Puterman, 1994; Sutton & Barto, 2018) at the token level. This MDP formulation facilitates the application of modern reinforcement learning (RL) algorithms (Schulman et al., 2017; 2015) to text generation. We then analyze reinforcement learning from human feedback (RLHF) (Ouyang et al., 2022) through the lens of reward maximization within this framework.

### 2.1. Token-level MDP for Text Generation

Following Ramamurthy et al. (2023), we model the text generation process as an MDP. The MDP is defined as a tuple $\mathcal{M} = \langle \mathcal{S}, \mathcal{A}, \mathcal{P}, r, \gamma, \rho, T \rangle$, where $\mathcal{S}$ is the state space and $\mathcal{A}$ is the action space, defined as the vocabulary of the language model. We use $\pi(a \mid s)$ to denote the policy (i.e., the LLM), which provides a distribution over actions given state $s$. At the beginning of generation, a prompt $\boldsymbol{x} = (x_1, x_2, \ldots, x_m)$ of length $m$ is sampled from the initial distribution $\rho(s)$ to form the initial state $s_0$. At each time step $t \in \{1, \ldots, T\}$, an action $a_t$ is sampled from the policy $\pi(\cdot \mid s_t)$. The MDP then transitions to the next state $s_{t+1}$ according to the transition function $\mathcal{P}$. The transition dynamics are deterministic, satisfying $\mathcal{P}(s_{t+1} \mid s_t, a_t) = 1$ if $s_{t+1} = s_t \oplus a_t$ (where $\oplus$ denotes concatenation), and 0 otherwise. The reward function $r(s_t, a_t)$ is provided at each step, and the model aims to maximize the discounted cumulative reward with discount factor $\gamma$. The episode

---

[1]Code available at `https://github.com/mansicer/REAR`.

terminates when an end-of-sequence token is generated or the maximum length $T$ is reached. We assume early-stopped sequences are padded to length $T$ for notational simplicity.

## 2.2. Reinforcement Learning from Human Feedback

The primary objective of RLHF is to find a policy $\pi(a \mid s)$ that maximizes the expected cumulative reward. Classical RLHF methods (Ouyang et al., 2022; Bai et al., 2022) typically train a model to maximize a reward function. The objective is formulated as:

$$\max_{\pi} \mathbb{E}_{\tau} \Big[ \sum_{t=0}^{T} \gamma^t \Big( r(s_t, a_t) - \beta D_{\mathrm{KL}} \big( \pi(\cdot|s_t) \| \pi_{\mathrm{ref}}(\cdot|s_t) \big) \Big) \Big], \tag{1}$$

where the trajectory $\tau = (s_0, a_0, \ldots, s_T, a_T)$ is generated by $\pi$, with $s_0 \sim \rho$, $a_t \sim \pi(\cdot \mid s_t)$, and $s_{t+1} \sim \mathcal{P}(s_t, a_t)$. Here, $D_{\mathrm{KL}}$ denotes the Kullback-Leibler divergence, and $\beta$ is a hyperparameter constraining the deviation of the learned policy $\pi$ from the reference policy $\pi_{\mathrm{ref}}$ (usually the initial base model). This objective can be reformulated within the framework of maximum entropy RL (Haarnoja et al., 2018; Li et al., 2025b) as follows.

**Proposition 2.1.** *The optimization problem in Equation* (1) *is equivalent to:*

$$\max_{\pi} \mathbb{E}_{s_0 \sim \rho} \big[ \mathbb{E}_{a_0 \sim \pi(\cdot|s_0)} \big[ Q^{\pi}(s_0, a_0) + \beta \mathcal{H}\big(\pi(\cdot|s_0)\big) \big] \big], \tag{2}$$

*where* $\mathcal{H}\big(\pi(\cdot|s_t)\big) = \mathbb{E}_{a_t \sim \pi(\cdot|s_t)} \left[ -\log \pi(a_t|s_t) \right]$ *is the entropy of* $\pi$ *at state* $s_t$, *and*

$$Q^{\pi}(s_0, a_0) = \mathbb{E}_{s_t \sim \mathcal{P}(s_0, a_0), a_t \sim \pi(\cdot|s_t)} \Big[ r_0(s_0, a_0) \\ + \sum_{t=1}^{T} \gamma^t \big( r_0(s_t, a_t) + \beta \mathcal{H}(\pi(\cdot|s_t)) \big) \Big] \tag{3}$$

*is the soft-Q function of policy* $\pi$. *The modified reward is defined as* $r_0(s, a) = r(s, a) + \beta \log \pi_{\mathrm{ref}}(a|s)$. *The soft-Q function* $Q^{\pi}$ *satisfies the Bellman equation:*

$$Q^{\pi}(s, a) = r_0(s, a) + \gamma \mathbb{E}_{\substack{s' \sim \mathcal{P}(s,a) \\ a' \sim \pi(\cdot|s')}} \Big[ Q^{\pi}(s', a') \\ + \beta \mathcal{H}\big(\pi(\cdot|s')\big) \Big] \tag{4} \\ = r_0(s, a) + \gamma V^{\pi}(s').$$

*We denote* $V^{\pi}(s) = \mathbb{E}_{a \sim \pi(\cdot|s)} \big[ Q^{\pi}(s, a) + \beta \mathcal{H}\big(\pi(\cdot|s)\big) \big]$ *as the value function of policy* $\pi$.

**Proposition 2.2.** *The optimal policy* $\pi^*$ *maximizing the objective in Equation* (2) *satisfies:*

$$\pi^*(a|s) = \exp \left( \frac{Q^{\pi^*}(s, a) - V^{\pi^*}(s)}{\beta} \right). \tag{5}$$

We defer the proof to Appendix A.1. Proposition 2.1 demonstrates that the RLHF objective can be interpreted within the maximum entropy RL framework, while Proposition 2.2 reveals the mapping between the optimal policy and value functions. As these value functions are defined upon the reward $r_0$, these results bridge the gap between RL-trained LLMs and their implicit reward structures (Li et al., 2025a). Given the prevalence of RL optimization in LLM research, these results provide a theoretical foundation for addressing the preference realignment problem.

# 3. Test-time Realignment via Reward Decomposition

This section establishes a theoretical framework for test-time preference realignment. We propose that the generation process can be guided by decomposing the underlying reward structure into question-related and preference-related components. Based on this formulation, we derive the *REAlignment Reward* (REAR), a tractable objective that enables dynamic control over preference intensity. Finally, we demonstrate how REAR serves as a scoring metric within test-time scaling (TTS) algorithms, such as best-of-$N$ sampling and DVTS, to facilitate scalable and aligned generation.

## 3.1. Reward Decomposition

In the context of preference alignment, the input state $s$ typically consists of a question and a generated sequence, while the preference description is denoted as $x_p$. To formulate the realignment problem, we first analyze the implicit objective of the pretrained LLM. Given that the pretrained policy $\pi(\cdot|s \oplus x_p)$ is trained on massive datasets, it is optimal with respect to a composite reward function under the Maximum Entropy RL framework. By contrasting the objectives conditioned with and without preference context $x_p$, we derive the following proposition regarding the latent reward structure.

**Proposition 3.1** (Reward Decomposition). *Let* $r_0(s, a)$ *be the question-related reward implicitly maximized by the base model* $\pi(a \mid s)$. *There exists a preference-related reward component* $r_p(s \oplus x_p, a)$ *such that the preference-conditioned policy* $\pi(a \mid s \oplus x_p)$ *maximizes the composite reward:*

$$r(s \oplus x_p, a) = r_0(s, a) + \alpha r_p(s \oplus x_p, a), \tag{6}$$

*where* $\alpha$ *is an inherent coefficient balancing the two reward components.*

The proof is provided in Appendix A.2. Proposition 3.1 offers a perspective for decomposing the preference alignment objective into two distinct reward components. The question-related reward term $r_0(s, a)$, which the original LLM maximizes, considers response quality solely in terms

of addressing the question. The preference-related reward, $r_p$, captures how well the generated response satisfies the preference $x_p$. $\alpha$ is an implicit coefficient, dependent on the LLM, that balances the trade-off between the question and the preference. Note that the subscript 0 in $r_0$ denotes the question-only base reward but is not an index over timesteps or contexts.

## 3.2. Realignment Reward (REAR)

The goal of realignment is to recalibrate the trade-off between the question-related reward $r_0$ and the preference-related reward $r_p$ during inference. We formulate this by defining the *REAlignment Reward* (REAR), which introduces a controllable coefficient $\hat{\alpha}$ to replace the inherent $\alpha$:

$$r_{\mathrm{REAR}}(s \oplus x_p, a) = r_0(s, a) + \hat{\alpha} r_p(s \oplus x_p, a). \quad (7)$$

where $\hat{\alpha}$ is a coefficient adjustable at test time. This flexibility allows us to decouple the implicit degree of preference alignment in the pretrained LLM. However, it is computationally intractable to evaluate $r_{\mathrm{REAR}}$ directly, as the individual reward terms are inaccessible. Fortunately, the maximum entropy RL framework (Haarnoja et al., 2018; Li et al., 2025a) allows us to express this reward in a computable form based on policy probabilities.

**Lemma 3.2.** *The realignment reward $r_{\mathrm{REAR}}(s \oplus x_p, a)$ is equal to*

$$
\begin{aligned}
r_{\mathrm{REAR}}(s \oplus x_p, a) = \frac{(\alpha - \hat{\alpha})\beta}{\alpha} \log \pi(a \mid s) \\
+ \frac{\hat{\alpha}\beta}{\alpha} \log \pi(a \mid s \oplus x_p) + Z(s) - \gamma Z(s'),
\end{aligned}
\quad (8)
$$

*where the state-dependent term $Z(s) = \left(1 - \frac{\hat{\alpha}}{\alpha}\right) V^\pi(s) + \frac{\hat{\alpha}}{\alpha} V^\pi(s \oplus x_p)$.*

We defer the detailed proof to Appendix A.3. Specifically, this reward term is equivalent to a linear combination of two log-probability terms plus state-dependent potential terms. In the context of potential-based reward shaping (Ng et al., 1999), $Z(s)$ serves as a potential function that does not alter policy optimality. Summing the token-level REAR over a trajectory $\tau = (s_0, a_0, s_1, a_1, \dots)$ yields the cumulative REAR:

$$
\begin{aligned}
R_{\mathrm{REAR}}(\tau) &= \sum_{s_t, a_t} \gamma^t r_{\mathrm{REAR}}(s_t \oplus x_p, a_t) \\
&= \sum_{s_t, a_t}^{\infty} \gamma^t \Big( (1 - \lambda)\beta \log \pi(a_t | s_t) \\
&\quad + \lambda\beta \log \pi(a_t | s_t \oplus x_p) \Big) + Z(s_0),
\end{aligned}
\quad (9)
$$

where we simplify the notation by defining $\lambda = \frac{\hat{\alpha}}{\alpha} > 0$. Intuitively, $\lambda > 1$ implies a higher weight on preference at test time compared to training, while $\lambda < 1$ reduces its importance. When $\lambda = 1$, the realignment is equivalent to direct model inference.

## 3.3. Test-time Scaling with REAR

Equation (9) provides a method to calculate realignment rewards for a trajectory segment, enabling the selection of outputs with the highest REAR values. Although Equation (9) cannot be computed directly due to the unknown terms $Z(s_0)$ and $\beta$, these terms are independent of the generated actions. Thus, ranking different responses relies solely on the following score function:

$$
\begin{aligned}
S_{\mathrm{REAR}}(\tau) = \sum_{t=0}^{T} \gamma^t \Big( (1 - \lambda) \log \pi(a_t | s_t) \\
+ \lambda \log \pi(a_t | s_t \oplus x_p) \Big).
\end{aligned}
\quad (10)
$$

Since Equation (10) can be computed using the log-probabilities of the base model, we can efficiently calculate the REAR score for each output during or after generation. Leveraging this flexibility, we integrate REAR into two test-time methods to boost performance.

**Best-of-$N$ sampling with REAR.** We sample $N$ responses and calculate the REAR score for each. The response with the highest $S_{\mathrm{REAR}}$ score is selected as the final output, according to Equation (10).

**Diverse Verifier Tree Search (DVTS) with REAR.** We employ the DVTS (Beeching et al., 2025) algorithm to select a final response, where the response is generated step-by-step via tree search and branches are selected based on REAR scores. DVTS extends standard tree search by explicitly promoting diversity among candidate trajectories, ensuring broader exploration. At each step, REAR scores guide expansion, balancing exploration with the exploitation of high-reward paths.

We defer algorithmic details to Appendix C. Compared to other test-time reward scoring methods (Lambert et al., 2025; Liu et al., 2024a; Zhang et al., 2025a; Mahan et al., 2024), REAR addresses preference alignment by solely rescaling inherent preferences, without requiring additional training or generation steps. This makes REAR highly flexible and deployment-friendly for common inference engines. Moreover, the REAR score is applicable to response segments, enabling the use of more complex search algorithms like DVTS, unlike general outcome-based rewards that only evaluate complete responses.

## 4. Related Work

**Preference Alignment**   Aligning LLMs with human preferences is a central challenge in AI safety and usability. Early and prominent approaches rely on training-based methods, particularly reinforcement learning from human feedback (RLHF) (Ouyang et al., 2022; Bai et al., 2022), where a reward model is trained on human preference data to fine-tune a base model. Subsequent work has sought to simplify this pipeline (Rafailov et al., 2023), bypassing the need for an explicit reward model. Other approaches focus on creating specialized data curricula (Zhang et al., 2025b) or maintaining original capabilities when adapting to new preferences (Li et al., 2023a; Wang et al., 2025; Li et al., 2025b). While effective, these training-based methods often require extensive data and are computationally expensive. This motivates a shift towards test-time alignment methods that adapt model behavior without updating weights. For instance, Zhang et al. (2025c) and Gao et al. (2024) propose techniques to modify the model's output distribution at each generation step to better align with given preferences. Our work builds on this line of research but focuses on scaling the alignment process through a novel reward formulation within a TTS framework rather than direct policy modification, which provides a stable and scalable performance improvement.

**Test-time Methods**   A growing body of work improves LLM behavior at inference time without weight updates, falling broadly into two families. *Decoding-time* methods modify the next-token distribution during autoregressive generation to steer outputs toward desired behaviors, including DExperts (Liu et al., 2021), Contrastive Decoding (Li et al., 2023b), and Drift (Kim et al., 2025). *Test-time scaling (TTS)* instead allocates additional compute through extended thinking (OpenAI, 2024; Guo et al., 2025; Muennighoff et al., 2025) or parallel search (Wang et al., 2024a; Comanici et al., 2025; Huang & Yang, 2025), and has been particularly successful in verifiable domains such as mathematics and coding (OpenAI, 2024; Zhang et al., 2026) via self-consistency (Wang et al., 2023; Li et al., 2024), process-based reward models (Lightman et al., 2024; Wang et al., 2024b), and search algorithms (Wei et al., 2022; Yao et al., 2023; Wang et al., 2024a). Extending TTS to open-ended preference alignment is harder because no simple verifier exists; generative reward models (Zhang et al., 2025a; Mahan et al., 2024; Liu et al., 2025) address this at the cost of efficiency and accuracy, while recent evidence (Li et al., 2025a) indicates the LLM is itself an implicit reward model. Unlike methods such as ARGS (Khanov et al., 2024) or IVG (Liu et al., 2024b) that train value heads, or speculative rejection (Sun et al., 2024a) that focuses on search efficiency given a reward model, REAR is a fully training-free reward formulation derived solely from the base model's internal proba-

bilities. REAR also resembles the decoding-time methods at the token level in that its reward takes a log-ratio form, but it differs in role: it is a trajectory-level scoring function used to rank complete or partial responses within TTS frameworks rather than a per-token decoding distribution, which makes it natively compatible with BoN and tree search such as DVTS. We provide a detailed point-by-point comparison with decoding-time methods in Appendix E.

## 5. Experiments

In this section, we evaluate the efficacy of REAR-guided test-time scaling (TTS) for preference alignment. We first describe the experimental setup in Section 5.1. We then compare our approach against baselines on preference alignment benchmarks and study robustness to long-context inputs in Section 5.2. Next, we examine the generalizability of REAR beyond preference alignment, including mathematical reasoning, multimodal tasks, and cross-family transfer to a non-Qwen base model, in Section 5.3. Finally, in Section 5.4, we analyze REAR by comparing its qualities with general reward models and examining sensitivity to the realignment coefficient $\lambda$.

### 5.1. Experimental Setup

**Baselines**   In addition to greedy decoding, we compare REAR against inference-time baselines from two categories. Implementation details are provided in Appendix C.

- **Test-time preference alignment methods**. We include two representative methods: Amulet (Zhang et al., 2025c) and Linear Alignment (LA) (Gao et al., 2024). These methods align generations with preferences by modifying the token-level generation probability distribution.

- **Test-time sampling with generative rewards**. We include a best-of-$N$ baseline scored by a generative reward model (GenRM) (Zhang et al., 2025a; Mahan et al., 2024), which uses the base model to produce a judgment for each candidate response. Because this approach does not provide reliable token- or segment-level scores, we apply it to best-of-$N$ (BoN) sampling but not to DVTS.

**Evaluation Benchmarks**   We evaluate preference alignment across three recent benchmarks that cover customized preference following and role-playing. The detailed benchmark descriptions and evaluation protocols can be found in Appendix D.

- **PrefEval** (Zhao et al., 2025) evaluates personalized responses in multi-turn conversations conditioned on

*Table 1.* Performance comparison of REAR-guided test-time methods and other baselines on various preference alignment benchmarks. Bold values indicate the best performance on the corresponding benchmark. Light blue indicates the method outperforms all baselines. Superscripts indicate the score difference compared to the greedy-decoding baseline on the same benchmark.

| Benchmark | DVTS w/ REAR (Ours) | BoN w/ REAR (Ours) | BoN w/ GenRM | Amulet | LA | Greedy |
|---|---|---|---|---|---|---|
| *PrefEval Scores* | | | | | | |
| Explicit Preference | $\mathbf{77.7}^{\uparrow 10.7}$ | $74.1^{\uparrow 7.1}$ | $69.0^{\uparrow 2.0}$ | $68.5^{\uparrow 1.5}$ | $64.2^{\downarrow 2.8}$ | 67.0 |
| Implicit Choice | $\mathbf{78.6}^{\uparrow 7.1}$ | $78.2^{\uparrow 6.7}$ | $74.7^{\uparrow 3.2}$ | $70.4^{\downarrow 1.1}$ | $78.0^{\uparrow 6.5}$ | 71.5 |
| Implicit Preference | $\mathbf{19.1}^{\uparrow 7.1}$ | $16.2^{\uparrow 4.2}$ | $12.9^{\uparrow 0.9}$ | $13.1^{\uparrow 1.1}$ | $12.8^{\uparrow 0.8}$ | 12.0 |
| Multifaceted Bench | $\mathbf{76.8}^{\uparrow 1.5}$ | $76.3^{\uparrow 1.0}$ | $76.1^{\uparrow 0.8}$ | $75.4^{\uparrow 0.1}$ | $75.6^{\uparrow 0.3}$ | 75.3 |
| Ping-Pong Bench | $3.03^{\uparrow 0.06}$ | $\mathbf{3.07}^{\uparrow 0.10}$ | $3.01^{\uparrow 0.04}$ | $2.87^{\downarrow 0.10}$ | $3.01^{\uparrow 0.04}$ | 2.97 |

*Table 2.* Long-context PrefEval scores across multiple conversation lengths. We report average LLM-evaluated scores for the explicit preference task and choice accuracy for the implicit choice task. Bold values indicate the best performance in each row. Light blue indicates the method outperforms all baselines. Superscripts indicate the score difference compared to the greedy-decoding baseline at the same context length.

| # Conversation Turns | 0 | 5 | 10 | 20 | 50 | 100 |
|---|---|---|---|---|---|---|
| Context Length (# Tokens) | 0 | 1k | 2k | 4k | 10k | 16k |
| **PrefEval Explicit Preference** | | | | | | |
| DVTS w/ REAR (Ours) | $\mathbf{77.7}^{\uparrow 10.7}$ | $\mathbf{33.4}^{\uparrow 20.1}$ | $\mathbf{20.7}^{\uparrow 12.2}$ | $\mathbf{16.6}^{\uparrow 9.9}$ | $\mathbf{8.7}^{\uparrow 4.5}$ | $\mathbf{7.2}^{\uparrow 3.2}$ |
| BoN w/ REAR (Ours) | $74.1^{\uparrow 7.1}$ | $29.9^{\uparrow 16.6}$ | $18.3^{\uparrow 9.8}$ | $14.8^{\uparrow 8.1}$ | $7.0^{\uparrow 2.8}$ | $5.6^{\uparrow 1.6}$ |
| BoN w/ GenRM | $69.0^{\uparrow 2.0}$ | $16.0^{\uparrow 2.7}$ | $10.3^{\uparrow 1.8}$ | $6.7^{0.0}$ | $4.4^{\uparrow 0.2}$ | $4.5^{\uparrow 0.5}$ |
| Greedy | 67.0 | 13.3 | 8.5 | 6.7 | 4.2 | 4.0 |
| **PrefEval Implicit Choice** | | | | | | |
| DVTS w/ REAR (Ours) | $\mathbf{78.6}^{\uparrow 7.1}$ | $52.5^{\uparrow 6.2}$ | $49.1^{\uparrow 6.8}$ | $\mathbf{45.1}^{\uparrow 9.1}$ | $\mathbf{47.0}^{\uparrow 8.0}$ | $\mathbf{42.5}^{\uparrow 4.1}$ |
| BoN w/ REAR (Ours) | $78.2^{\uparrow 6.7}$ | $\mathbf{52.6}^{\uparrow 6.3}$ | $\mathbf{50.1}^{\uparrow 7.8}$ | $45.0^{\uparrow 9.0}$ | $46.2^{\uparrow 7.2}$ | $41.7^{\uparrow 3.3}$ |
| BoN w/ GenRM | $74.7^{\uparrow 3.2}$ | $47.0^{\uparrow 0.7}$ | $45.6^{\uparrow 3.3}$ | $39.3^{\uparrow 3.3}$ | $40.7^{\uparrow 1.7}$ | $37.7^{\downarrow 0.7}$ |
| Greedy | 71.5 | 46.3 | 42.3 | 36.0 | 39.0 | 38.4 |

previously stated user preferences. It tests a model's ability to infer, retain, and apply preference information. PrefEval includes three subsets: explicit preference, implicit choice, and implicit preference.

- **Multifaceted Bench** (Lee et al., 2024) evaluates whether an LLM generates context-specific responses tailored to user preferences. Each instance includes a synthetic system message and reference answers, along with rubric-based evaluation criteria.

- **Ping-Pong** (Gusev, 2024) evaluates role-playing in multi-turn conversations. Since role-playing can be viewed as preference alignment to a persona and interaction style, we use this benchmark to assess performance in a practical setting.

We use the vLLM inference engine (Kwon et al., 2023) for response generation and keep sampling hyperparameters consistent across our implementations. For Amulet and LA, we use the authors' reference implementation (Zhang et al.,

2025c). Unless otherwise specified, we use $N = 16$ samples for BoN and an approximately matched compute budget for DVTS. Further details are provided in Appendix C.

### 5.2. Performance on Preference Alignment Benchmarks

We compare our methods against baselines on PrefEval, Multifaceted Bench, and Ping-Pong, using Qwen2.5-7B-Instruct (Yang et al., 2024) as the base model. As shown in Table 1, both BoN with REAR and DVTS with REAR outperform the baselines on most benchmarks. To summarize the overall gains, we compute an average over the five reported metrics in Table 1. Under this aggregate metric, DVTS with REAR improves over greedy decoding, BoN with GenRM, Amulet, and LA by 11.6%, 8.3%, 10.8%, and 9.3%, respectively. BoN with REAR improves over the same baselines by 8.4%, 5.2%, 7.6%, and 6.1%, respectively. GenRM yields only limited gains, suggesting that the base model does not reliably verify its own responses under our prompting scheme. Test-time preference alignment methods, including Amulet and LA, also underperform on

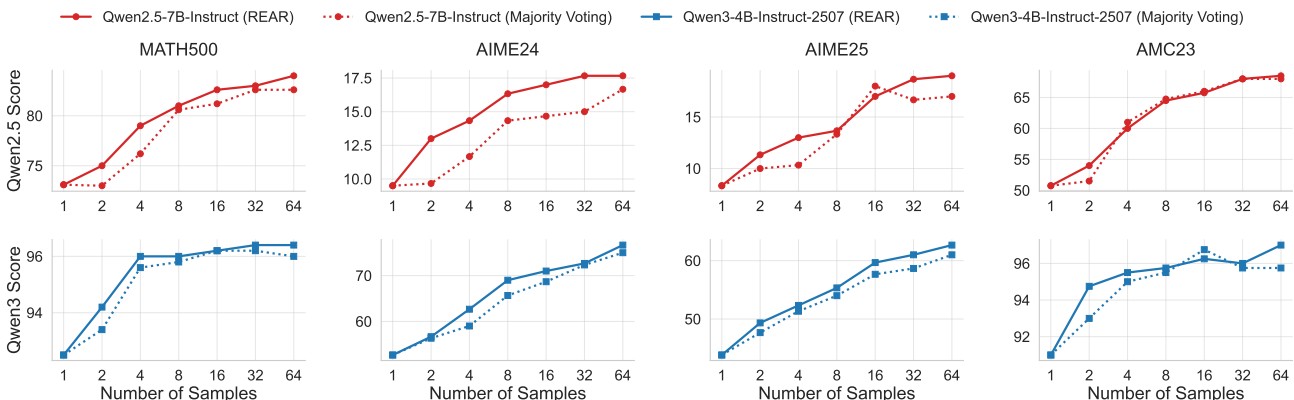

*Figure 2.* The scores of answer selection with REAR and majority voting on mathematical tasks including MATH500, AIME24, AIME25, and AMC23. We show the results of Qwen-2.5-7B-Instruct (top) and Qwen3-4B-Instruct-2507 (bottom) with up to $N = 64$ samples.

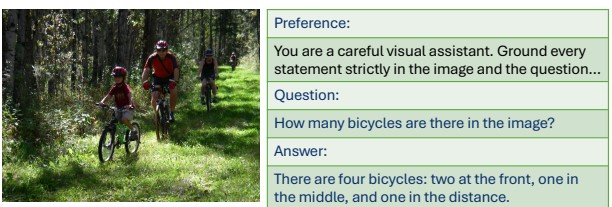

*Figure 3.* An example of the MMHal-Bench, where we inject text-based preference information to the task input for REAR.

*Table 3.* Comparisons of scores and hallucination rates on MMHal-Bench with the Qwen3-VL-8B-Instruct model.

| Method | MMHal Score (↑) | Hallucination (↓) |
|---|---|---|
| BoN w/ REAR | **84.20** | **21.87** |
| BoN w/ GenRM | 80.21 | 23.96 |
| Greedy | 78.99 | 28.12 |

these benchmarks.

Across benchmarks, we observe a substantial performance drop on PrefEval implicit preference relative to the other two PrefEval subsets, consistent with prior work (Zhao et al., 2025). Our methods also perform well on the highly customized Multifaceted Bench, which uses instance-specific rubrics. On Ping-Pong, which emphasizes multi-turn role-playing, BoN performs better than DVTS. We hypothesize that DVTS's step-wise exploration encourages diverse continuations that can drift from the intended persona in long conversations. Since our sampling-based methods naturally scale with the sampling budget, we also report scaling curves in Appendix F.

**Robustness on Long-context Input** Since REAR is derived from the base model's own log-probabilities, it can be more robust under distribution shifts. We evaluate long-context robustness by augmenting PrefEval conversations with additional turns inserted between the preference context and the question, following Zhao et al. (2025). As shown in Table 2, our methods consistently outperform the baselines across context lengths. We exclude Amulet and LA due to out-of-memory failures on long-context inputs. In contrast, best-of-$N$ with GenRM degrades substantially as context length increases, likely because the augmented input disrupts the model's capability in judging responses.

### 5.3. Generalizability of REAR

In this section, we investigate whether REAR-guided search can generalize to more general LLM tasks. Beyond tasks that naturally come with bespoke preference descriptions, REAR can also improve tasks where preferences are not explicitly provided by introducing a task-specific preference to guide the text generation process. We validate this generalizability along three axes: mathematical reasoning, visual hallucination, and cross-family transfer to a non-Qwen base model, showing that REAR yields consistent test-time scaling gains when the preference is appropriately specified.

**Mathematical Problems** Although REAR is designed for preference alignment with explicit preference text, the same formulation applies whenever the task can be specified via an additional instruction or constraint. We assess this hypothesis on mathematical reasoning, where each problem has a deterministic final answer that enables automatic verification (Hendrycks et al., 2021). We consider MATH500, a commonly used 500-problem subset of the MATH test set, and three competition-style benchmarks including AIME24, AIME25, and AMC23. These benchmarks are well-suited for test-time scaling because correctness can be evaluated via exact-match on the extracted final answer, making sampling-based selection particularly effective. We therefore introduce majority voting (Wang et al., 2023) as a baseline, which selects the most frequent answer among sampled candidates. To validate whether the out-

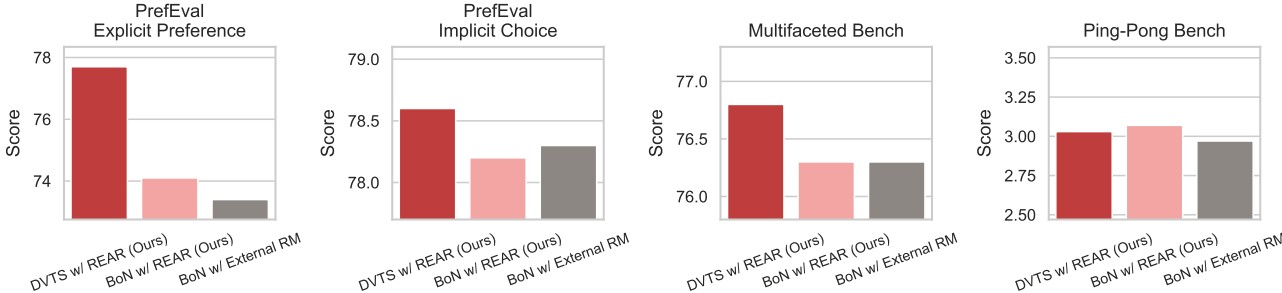

*Figure 4.* Comparison between REAR-guided methods and an external reward model baseline. We compare our methods (BoN and DVTS with REAR) against best-of-$N$ sampling using the Skywork-Reward-Llama-8B model (Liu et al., 2024a).

*Table 4.* Cross-family generalization on Llama-3.1-8B-Instruct with $\lambda = 20$. REAR-guided BoN and DVTS outperform Greedy decoding and BoN with GenRM on all three preference benchmarks, confirming that the reward decomposition is not specific to the Qwen family. Bold values indicate the best performance on the corresponding benchmark. Light blue indicates the method outperforms all baselines. Superscripts indicate the score difference compared to the greedy-decoding baseline on the same benchmark.

| Benchmark | DVTS w/ REAR (Ours) | BoN w/ REAR (Ours) | BoN w/ GenRM | Greedy |
|---|---|---|---|---|
| PrefEval Choice | $\mathbf{82.1}^{\uparrow 5.7}$ | $79.5^{\uparrow 3.1}$ | $77.9^{\uparrow 1.5}$ | 76.4 |
| PrefEval Explicit | $\mathbf{86.4}^{\uparrow 11.8}$ | $78.4^{\uparrow 3.8}$ | $74.6^{0.0}$ | 74.6 |
| Multifaceted Bench | $\mathbf{74.6}^{\uparrow 1.1}$ | $74.3^{\uparrow 0.8}$ | $73.6^{\uparrow 0.1}$ | 73.5 |

put chosen with REAR can be better than majority voting, we select the candidate with the most accumulated REAR scores. In the experiments, we test two language models including Qwen2.5-7B-Instruct and Qwen3-4B-Instruct-2507. As shown in Figure 2, REAR generally outperforms majority voting in these four mathematical tasks, indicating that REAR can provide consistent test-time scaling gains across models when the preference text is appropriately specified. We further study how sensitive REAR is to the exact wording of this preference text in Appendix G, finding that the gains are robust to rephrasing a task-relevant preference but vanish for an unrelated one, confirming that REAR exploits genuine alignment between the preference and the task rather than acting as a generic generation prior.

**Visual Hallucination Problems**    We further apply REAR to multimodal models for visual hallucination detection using *MMHal-Bench* (Sun et al., 2024b), which probes whether a model's response is grounded in the input image, as shown in Figure 3. We adopt the benchmark's official evaluation protocol and report both the overall score and the hallucination rate, both of which are computed by the benchmark's LLM-as-a-judge pipeline. In this experiment, we use Qwen3-VL-8B-Instruct (Bai et al., 2025) as the base model. When injecting preference information, we place the preference in the system prompt and provide the image and corresponding question in the user prompt. We use a

preference text that emphasizes factual grounding and calibrated uncertainty. As shown in Table 3, BoN with REAR improves the overall score and reduces the hallucination rate compared to greedy decoding and BoN with GenRM. We report the detailed implementation in Appendix C.

**Cross-Model Generalization**    The decomposition underlying REAR (Proposition 3.1) only assumes a well-aligned base policy and is not specific to any particular model family. To verify this empirically, we apply REAR to Llama-3.1-8B-Instruct (Grattafiori et al., 2024) on three preference benchmarks, reusing the same $\lambda = 20$ used for the Qwen experiments without any per-task or per-model retuning. As shown in Table 4, both BoN with REAR and DVTS with REAR outperform greedy decoding and BoN with GenRM, with DVTS reaching the highest score on every benchmark. This indicates that REAR's gains transfer across model families and that $\lambda = 20$ remains a sensible default beyond the Qwen models used elsewhere in our evaluation.

### 5.4. Analysis on Realignment Rewards

**Comparisons with Reward Models**    REAR provides a scoring signal that is conceptually similar to dedicated reward models for evaluating responses. We therefore compare REAR to an external reward model (RM) on preference alignment benchmarks. Specifically, we use best-of-$N$ sampling scored by Skywork-Reward-Llama-8B (Liu et al.,

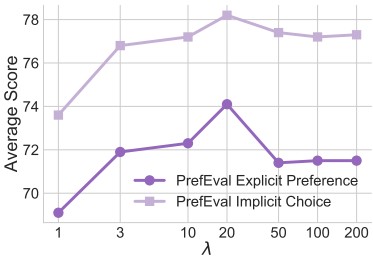

*Figure 5.* Similar trends on the benchmark scores of best-of-$N$ sampling with REAR are found on PrefEval explicit preference and implicit choice data with multiple $\lambda$ choices.

2024a), which performs strongly on RewardBench (Lambert et al., 2025) and is comparable in size to our base model. Tree search methods such as DVTS are not directly applicable to external RMs because they typically do not provide reliable segment-level scores for partial continuations, whereas REAR can score trajectory segments via token-level log-probabilities. As shown in Figure 4, REAR-guided methods consistently outperform best-of-$N$ with the external RM. Under the same best-of-$N$ strategy, REAR outperforms the external RM on three of four tasks, suggesting that REAR provides competitive self-evaluation signals despite not training additional models on additional data. Because REAR computes its score from the base model's own log-probabilities, it also avoids the overhead of loading and running a separate reward model; we report the resulting latency and throughput gains over external-RM and GenRM baselines in Appendix I.

**Performance on Different $\lambda$ Choices** The coefficient $\lambda$ controls the strength of realignment by scaling the preference-related component. Larger values of $\lambda$ place greater emphasis on preference satisfaction, whereas smaller values prioritize answering the question and may insufficiently enforce preferences. We evaluate both BoN and DVTS with REAR across a range of $\lambda$ values; the results are reported in Appendix H. Overall, $\lambda = 20$ yields stable and strong performance across most tasks. In addition, we report the BoN results on PrefEval explicit preference and implicit choice tasks in Figure 5. On both tasks, performance first improves and then degrades as $\lambda$ increases, suggesting that optimal generations require an appropriate trade-off between answering questions and adhering to preferences. The variation across $\lambda \in \{3, 10, 20, 50\}$ is small relative to the gap between REAR and baselines, so the headline gains in Tables 1 and 2 are not an artifact of per-task $\lambda$ tuning. To further rule out test-set tuning, we conduct a held-out validation sweep that selects $\lambda$ on a validation split and reports the test-split score; the procedure re-selects $\lambda = 20$ in nearly all task-method configurations (Table 6), confirming that $\lambda = 20$ is a robust default rather than a test-set-tuned value.

# 6. Conclusion

In this work, we introduced the REAlignment Reward (REAR), a novel and efficient reward that realigns LLMs to user preferences at test time. By decomposing the underlying reward into question-related and preference-related components, we can calculate REAR directly from the model's own policy probabilities. We further integrate two test-time scaling methods, best-of-$N$ sampling and DVTS, with REAR, enabling effective preference realignment without training. Extensive experiments show that REAR-guided TTS methods significantly outperform recent test-time baselines across a range of preference-alignment benchmarks. Our work provides a scalable and general solution for general LLM generation and enables test-time scaling to more subjective, open-ended domains without additional models.

**Limitations** Despite these promising results, our work has several limitations. First, like any TTS method with a single tunable knob, REAR exposes the realignment strength $\lambda$ to the user. We find $\lambda = 20$ to be a robust default in our experiments, and validation-based selection shows its stability, so per-task tuning may not be required in practice. However, settings with substantially different preference distributions may still benefit from a small validation sweep. Second, while TTS is more lightweight than fine-tuning, identifying the sweet spot of REAR-guided TTS methods without incurring excessive computational cost remains a promising direction for future work.

# Acknowledgements

This research is supported by the Ministry of Education, Singapore, under its MOE AcRF Tier 2 Award MOE-T2EP20223-0003. Any opinions, findings and conclusions or recommendations expressed in this material are those of the author(s) and do not reflect the views of the Ministry of Education, Singapore. We thank Skywork AI for providing part of the computational resources.

# Impact Statement

This paper presents work whose goal is to advance the field of Machine Learning. There are many potential societal consequences of our work, none which we feel must be specifically highlighted here.

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

# A. Deferred Proofs

## A.1. Proof of Proposition 2.1 and Proposition 2.2

The objective of RLHF is to maximize the expected discounted reward regularized by the KL divergence between the learned policy $\pi$ and a reference policy $\pi_{\text{ref}}$:

$$\max_{\pi} \mathbb{E}_{\tau \sim \pi} \left[ \sum_{t=0}^{T} \gamma^t \left( r(s_t, a_t) - \beta D_{\text{KL}} \left( \pi(\cdot|s_t) \| \pi_{\text{ref}}(\cdot|s_t) \right) \right) \right], \tag{11}$$

where the expectation is over trajectories $\tau = (s_0, a_0, s_1, \dots)$ sampled from the policy $\pi$.

First, we expand the KL divergence term:

$$D_{\text{KL}} \left( \pi(\cdot|s_t) \| \pi_{\text{ref}}(\cdot|s_t) \right) = \mathbb{E}_{a_t \sim \pi(\cdot|s_t)} \left[ \log \pi(a_t|s_t) - \log \pi_{\text{ref}}(a_t|s_t) \right]. \tag{12}$$

Substituting this into the objective and taking the expectation over actions inside the summation gives:

$$\max_{\pi} \mathbb{E}_{\tau \sim \pi} \left[ \sum_{t=0}^{T} \gamma^t \left( r(s_t, a_t) - \beta \left( \log \pi(a_t|s_t) - \log \pi_{\text{ref}}(a_t|s_t) \right) \right) \right]. \tag{13}$$

We can rearrange the terms within the summation:

$$\max_{\pi} \mathbb{E}_{\tau \sim \pi} \left[ \sum_{t=0}^{T} \gamma^t \left( \left( r(s_t, a_t) + \beta \log \pi_{\text{ref}}(a_t|s_t) \right) - \beta \log \pi(a_t|s_t) \right) \right]. \tag{14}$$

Let us define a reshaped reward function $r_0(s_t, a_t) = r(s_t, a_t) + \beta \log \pi_{\text{ref}}(a_t|s_t)$. Additionally, we recognize that the term $-\mathbb{E}_{a_t \sim \pi(\cdot|s_t)} \left[ \log \pi(a_t|s_t) \right]$ is the entropy of the policy, denoted by $\mathcal{H}\left( \pi(\cdot|s_t) \right)$. With these substitutions, the objective becomes:

$$\max_{\pi} \mathbb{E}_{\tau \sim \pi} \left[ \sum_{t=0}^{T} \gamma^t \left( r_0(s_t, a_t) + \beta \mathcal{H}\left( \pi(\cdot|s_t) \right) \right) \right]. \tag{15}$$

This is the standard objective for maximum entropy reinforcement learning. The expected return in this framework is the definition of the soft value function $V^\pi(s_0)$. The objective can thus be written in terms of the soft Q-function and entropy at the initial state, which is equivalent to the formulation in Equation (2).

Now we proceed to prove Proposition 2.2. At any state $s$, the optimal policy $\pi^*$ maximizes the soft value function:

$$V^{\pi^*}(s) = \max_{\pi} \mathbb{E}_{a \sim \pi(\cdot|s)} \left[ Q^{\pi^*}(s, a) - \beta \log \pi(a|s) \right]. \tag{16}$$

This is a constrained optimization problem where $\sum_a \pi(a|s) = 1$. We form the Lagrangian:

$$\mathcal{L}(\pi, \zeta) = \sum_a \pi(a|s) \left( Q^{\pi^*}(s, a) - \beta \log \pi(a|s) \right) - \zeta \left( \sum_a \pi(a|s) - 1 \right). \tag{17}$$

Taking the derivative with respect to $\pi(a|s)$ and setting it to zero yields:

$$Q^{\pi^*}(s, a) - \beta(\log \pi(a|s) + 1) - \zeta = 0. \tag{18}$$

Solving for $\pi(a|s)$, we get:

$$\pi(a|s) = \exp \left( \frac{Q^{\pi^*}(s, a) - \zeta}{\beta} - 1 \right) \propto \exp \left( \frac{Q^{\pi^*}(s, a)}{\beta} \right). \tag{19}$$

Using the normalization constraint $\sum_a \pi(a|s) = 1$, the partition function is $Z(s) = \sum_a \exp(Q^{\pi^*}(s, a)/\beta)$. Substituting $\pi^*$ back into the definition of $V^{\pi^*}(s)$:

$$V^{\pi^*}(s) = \sum_a \pi^*(a|s) \left( Q^{\pi^*}(s, a) - \beta \left( \frac{Q^{\pi^*}(s, a)}{\beta} - \log Z(s) \right) \right) = \beta \log Z(s). \tag{20}$$

Thus, $\log Z(s) = V^{\pi^*}(s)/\beta$. The optimal policy can be written as:

$$\pi^*(a|s) = \frac{\exp(Q^{\pi^*}(s,a)/\beta)}{Z(s)} = \exp\left(\frac{Q^{\pi^*}(s,a) - V^{\pi^*}(s)}{\beta}\right). \tag{21}$$

This concludes the proof.

### A.2. Proof of Proposition 3.1

According to Proposition 2.2, for any policy $\pi$ that is optimal with respect to a reward function $r$ in the maximum entropy framework, we have the relation:

$$r(s,a) = \beta \log \pi(a|s) + V^{\pi}(s) - \gamma V^{\pi}(s'). \tag{22}$$

Applying this to the base model $\pi(a|s)$ which optimizes $r_0(s,a)$:

$$r_0(s,a) = \beta \log \pi(a|s) + V^{\pi}(s) - \gamma V^{\pi}(s'). \tag{23}$$

Similarly, applying this to the preference-conditioned model $\pi(a|s \oplus x_p)$ which optimizes $r(s \oplus x_p, a)$:

$$r(s \oplus x_p, a) = \beta \log \pi(a|s \oplus x_p) + V^{\pi_p}(s \oplus x_p) - \gamma V^{\pi_p}(s' \oplus x_p), \tag{24}$$

where $\pi_p(\cdot) = \pi(\cdot|s \oplus x_p)$ and $V^{\pi_p}$ is its corresponding value function. We can simply define the preference-related reward $r_p$ as the scaled difference:

$$r_p(s \oplus x_p, a) = \frac{1}{\alpha}\left(r(s \oplus x_p, a) - r_0(s,a)\right). \tag{25}$$

Substituting the expressions above, we obtain:

$$r_p(s \oplus x_p, a) = \frac{\beta}{\alpha} \log \frac{\pi(a|s \oplus x_p)}{\pi(a|s)} + \frac{1}{\alpha}\left(\Delta V(s) - \gamma \Delta V(s')\right), \tag{26}$$

where $\Delta V(s) = V^{\pi_p}(s \oplus x_p) - V^{\pi}(s)$. Since potential-based shaping terms do not affect the optimal policy, the core preference signal is captured by the log-probability ratio. This demonstrates the existence of such a reward function satisfying the decomposition.

### A.3. Proof of Lemma 3.2

From Equation (7), the realignment reward is defined as:

$$r_{\text{REAR}}(s \oplus x_p, a) = r_0(s,a) + \hat{\alpha} r_p(s \oplus x_p, a). \tag{27}$$

In the proof of Proposition 3.1, we established that:

$$r_p(s \oplus x_p, a) = \frac{\beta}{\alpha}\left(\log \pi(a|s \oplus x_p) - \log \pi(a|s)\right) + \frac{1}{\alpha}\left(\Delta V(s) - \gamma \Delta V(s')\right), \tag{28}$$

where $\Delta V(s) = V^{\pi}(s \oplus x_p) - V^{\pi}(s)$. Also, from the Bellman equation for the base policy $\pi(\cdot|s)$ maximizing $r_0$, we have:

$$r_0(s,a) = \beta \log \pi(a|s) + V^{\pi}(s) - \gamma V^{\pi}(s'). \tag{29}$$

Substituting these into the REAR definition:

$$\begin{aligned}
r_{\text{REAR}}(s \oplus x_p, a) &= \beta \log \pi(a|s) + V^{\pi}(s) - \gamma V^{\pi}(s') \\
&\quad + \hat{\alpha}\left[\frac{\beta}{\alpha}\left(\log \pi(a|s \oplus x_p) - \log \pi(a|s)\right) + \frac{1}{\alpha}\left(\Delta V(s) - \gamma \Delta V(s')\right)\right] \\
&= \left(1 - \frac{\hat{\alpha}}{\alpha}\right)\beta \log \pi(a|s) + \frac{\hat{\alpha}\beta}{\alpha} \log \pi(a|s \oplus x_p) + Z(s) - \gamma Z(s'),
\end{aligned} \tag{30}$$

where we define $Z(s)$ by collecting the state-dependent terms:

$$Z(s) = V^{\pi}(s) + \frac{\hat{\alpha}}{\alpha}\Delta V(s) = V^{\pi}(s) + \frac{\hat{\alpha}}{\alpha}\left(V^{\pi}(s \oplus x_p) - V^{\pi}(s)\right) = \left(1 - \frac{\hat{\alpha}}{\alpha}\right)V^{\pi}(s) + \frac{\hat{\alpha}}{\alpha}V^{\pi}(s \oplus x_p). \tag{31}$$

This confirms the expression in the lemma. The term $Z(s) - \gamma Z(s')$ acts as a potential-based reward shaping term, which does not alter the optimal policy.

---

**Algorithm 1** Best-of-$N$ sampling with REAR

---

**Input:** Prompt $s_0$, Preference $x_p$, Model $\pi$, Number of samples $N$, realignment strength $\lambda$.
Initialize candidate set $\mathcal{C} \leftarrow \emptyset$
**for** $i = 1$ to $N$ **do**
    Sample trajectory $\tau_i \sim \pi(\cdot|s_0 \oplus x_p)$
    Calculate score $S_i \leftarrow S_{\text{REAR}}(\tau_i)$ using Equation (10)
    $\mathcal{C} \leftarrow \mathcal{C} \cup \{(\tau_i, S_i)\}$
**end for**
**Return** $\tau^*$ with the highest score in $\mathcal{C}$

---

## B. Declaration on the Use of LLMs

We acknowledge the use of Large Language Models (LLMs) to assist in the preparation of this manuscript. Specifically, LLMs were utilized for the following tasks: (1) generating boilerplate code for experiment scripts, (2) assisting with the implementation of baselines and plotting scripts for visualizing results, (3) performing grammar and spelling checks to improve readability, and (4) proofreading the manuscript for clarity and correctness. All content, including the final text, figures, and scientific contributions, were curated and verified by the authors.

## C. Implementation Details

Our experiments use a framework based on the vLLM inference engine (Kwon et al., 2023). For all methods, we serve a suite of post-trained, instruction-following models through the same inference stack to ensure consistent and efficient generation, including Qwen2.5-7B-Instruct, Qwen3-4B-Instruct-2507, Qwen3-VL-8B-Instruct, and Llama-3.1-8B-Instruct. We choose these models because of their broad adoption and moderate baseline performance on our benchmarks, which leaves headroom for test-time scaling methods.

**Calculation of REAR Scores** We compute the REAR score using token-level log-probabilities under two contexts: (i) the full prompt including preference information and (ii) the prompt containing only the question. We use SGLang frontend APIs to obtain log-probabilities for each response token. Log-probabilities under the full prompt are produced during generation, whereas log-probabilities under the question-only prompt are obtained via an additional forward pass over the concatenation of the question-only prompt and the generated response. We then combine the two log-probability sequences as a weighted sum controlled by $\lambda$, following our formulation. For both complete and partial responses, we set the discount factor $\gamma = 1$ to weight all tokens equally.

**TTS Methods** We adapt our REAR scores to two TTS methods, best-of-$N$ sampling (BoN) (Stiennon et al., 2020) (Algorithm 1) and dynamic verifier tree search (DVTS) (Beeching et al., 2025) (Algorithm 2). For BoN, we directly use the inference engine to generate multiple responses in separate requests, and then select the response with the highest REAR score. For DVTS, we use the line break as the delimiter of each tree search step, where the algorithm selects the expanded branch of each node according to the REAR score. In our experiments, unless specified, we set the number of samples to 16 for all BoN methods including the baselines. For DVTS, we set an equivalent compute budget to the BoN method by setting its expansion width and initial tree nodes both to 4. According to Beeching et al. (2025), this setting is comparable to the $N = 16$ setting for BoN. All generated responses are sampled using a temperature of 1.0 and the maximum generated length is set to 2048 tokens. Unless otherwise stated, we set the realignment strength to $\lambda = 20$ for all benchmarks and report a full sweep over $\lambda \in \{3, 10, 20, 50\}$ in Appendix H.

**Amulet and Linear Alignment (LA)** We use the implementation of Amulet and LA provided by the Amulet paper (Zhang et al., 2025c) to run the experiments[2]. We do not change the default hyper-parameters of these baselines. For Amulet, experiments are run with an iteration number of 60 for test-time alignment.

**Best-of-$N$ with Generative RM (GenRM)** This baseline leverages the base model as its own judge. Each generated response is appended with a template that prompts the model to evaluate whether the response is preferred. The final reward

---

[2]https://github.com/zowiezhang/Amulet

**Algorithm 2** DVTS with REAR

**Input:** Prompt $s_0$, Preference $x_p$, Model $\pi$, Width $W$, Depth $T$, realignment strength $\lambda$.
Initialize active set $A_0 \leftarrow \{s_0\}$
**for** $t = 1$ to $T$ **do**
   Initialize candidate set $C_t \leftarrow \emptyset$
   **for** each trajectory $\tau \in A_{t-1}$ **do**
      Generate $W$ continuations from $\pi(\cdot | \tau \oplus x_p)$
      Calculate REAR score for each continuation using Equation (10)
      Add continuations to $C_t$
   **end for**
   Select $W$ diverse trajectories $A_t \subset C_t$ based on REAR scores and diversity
**end for**
**Return** trajectory in $A_T$ with highest REAR score

*Table 5.* Ablation studies on the realignment strength $\lambda$ for REAR across preference-alignment, role-playing, and mathematical benchmarks.

*(a)* $\lambda$ ablation for REAR on PrefEval and Multifaceted benchmarks.

| Method | $\lambda$ | | | |
|---|---|---|---|---|
| | 3 | 10 | 20 | 50 |
| *PrefEval Explicit Preference* | | | | |
| BoN w/ REAR | 71.9 | 72.3 | **74.1** | 71.4 |
| DVTS w/ REAR | **77.4** | 76.4 | 76.3 | 75.1 |
| *PrefEval Implicit Choice* | | | | |
| BoN w/ REAR | 76.8 | 77.2 | **78.2** | 77.4 |
| DVTS w/ REAR | 73.8 | 76.2 | **78.6** | 77.4 |
| *PrefEval Implicit Preference* | | | | |
| BoN w/ REAR | 14.6 | 15.1 | 15.4 | **16.2** |
| DVTS w/ REAR | 14.7 | 17.4 | **19.1** | 18.1 |
| *Multifaceted Bench* | | | | |
| BoN w/ REAR | 75.4 | 76.0 | **76.3** | 75.3 |
| DVTS w/ REAR | 74.5 | 75.3 | **76.8** | 75.6 |

*(b)* $\lambda$ ablation for REAR on Ping-Pong Bench.

| BoN w/ REAR | | | | |
|---|---|---|---|---|
| $\lambda$ | 3 | 10 | 20 | 50 |
| Score | 2.92 | 2.97 | 3.02 | **3.07** |

| DVTS w/ REAR | | | | |
|---|---|---|---|---|
| $\lambda$ | 0.3 | 0.5 | 1 | 1.5 |
| Score | 2.88 | 2.99 | **3.03** | 2.87 |

*(c)* $\lambda$ ablation for REAR on mathematical benchmarks.

| Task | $\lambda$ | | | |
|---|---|---|---|---|
| | 3 | 10 | 20 | 50 |
| MATH500 | 82.4 | **82.6** | **82.6** | 82.2 |
| AIME25 | 16.7 | 16.7 | **17.0** | 16.0 |
| AMC23 | **65.8** | 65.3 | **65.8** | 65.0 |

is calculated from the log-probability difference between the model generating "Yes" and "No". We defer the prompt templates of the GenRM baseline to Appendix J.

**Best-of-$N$ with External RM** This approach uses an external, dedicated reward model, Skywork-Reward-Llama-8B (Liu et al., 2024a), hosted on an independent inference endpoint. For each candidate response, the prompt and the response are sent to this external model, which returns a scalar reward score.

## D. Benchmarking Tasks and Evaluation Protocols

In this section, we provide a detailed description of the evaluation protocols used for each benchmark in our experiments. For benchmarks with verifiable ground-truth answers (e.g., PrefEval implicit choice and mathematical reasoning), we report accuracy computed from the extracted final answer. For open-ended generation benchmarks, we adopt LLM-as-a-judge evaluation and use GPT-4.1 as the judge via the OpenAI API.

**PrefEval** The PrefEval benchmark (Zhao et al., 2025) is evaluated across its three distinct data types, each with a specific protocol. For **explicit preference**, the task is evaluated using an LLM-as-a-judge. For each generated response, a series of automated checks assesses different aspects of quality and preference alignment, including helpfulness, preference violation,

*Table 6.* Validation-based $\lambda$ selection on Qwen2.5-7B-Instruct. For each (method, task) cell we sweep $\lambda \in \{3, 10, 20, 50\}$ on a 20% held-out validation split and report the test-split score using the validation-selected $\lambda$ alongside the $\lambda$ used in the main tables.

| Method | Task | Val-best $\lambda$ (test score) | Main-table $\lambda$ (test score) |
|--------|------|------------------------------|----------------------------------|
| BoN | PrefEval Explicit | 20 (73.2) | 20 (73.2) |
| BoN | PrefEval Choice | 20 (79.0) | 20 (79.0) |
| BoN | PrefEval Implicit | 20 (16.9) | 50 (17.1) |
| BoN | Multifaceted | 20 (76.7) | 20 (76.7) |
| DVTS | PrefEval Explicit | 3 (76.5) | 3 (76.5) |
| DVTS | PrefEval Choice | 20 (78.2) | 20 (78.2) |
| DVTS | PrefEval Implicit | 20 (18.8) | 20 (18.8) |
| DVTS | Multifaceted | 20 (76.8) | 20 (76.8) |

*Table 7.* Comparison between REAR and three decoding-time methods most often cited as related work.

| Aspect | DExperts | CD | Drift | REAR (Ours) |
|--------|----------|----|----|-------------|
| Role | Per-token decoding | Per-token decoding | Per-token decoding | Trajectory reward |
| TTS | No | No | No | Yes (BoN + DVTS) |
| Require | Trained expert model | Two sized models | Attributes + preference pairs | Preference text only |

consistency, and hallucination. The final score is an aggregated metric that reflects overall preference-following accuracy. The evaluation protocol for **implicit preference** is identical to that of explicit preference, using the same LLM-as-a-judge and the same set of automated checks. For **implicit choice**, this is a multiple-choice task where the model must select the best response from four options. The evaluation protocol extracts the model's choice from its generated output and compares it to the ground-truth correct answer. The final performance is measured by accuracy.

**Multifaceted Bench**    For the Multifaceted Bench (Lee et al., 2024), we also employ an LLM-as-a-judge for evaluation. The judge assesses the model's generated response based on a set of rubrics that are provided within each data sample. It assigns a score from 1 to 5 for each rubric. The final reported score is the average of these scores across all rubrics.

**Ping-Pong Bench**    The Ping-Pong benchmark (Gusev, 2024) for role-playing is evaluated using an LLM-as-a-judge. The judge evaluates the entire conversation using three criteria: **stay-in-character score** (persona consistency), **entertaining score** (engagement), and **fluency score** (language quality). The final metric is the average across criteria. We evaluate on the English version of Ping-Pong-v2. Unlike the original benchmark, which uses gpt-4o-mini as the interrogator model to generate multi-turn user messages, we use the same base model (Qwen2.5-7B-Instruct) as the interrogator to reduce API costs. This choice may slightly reduce absolute scores compared to the original setting; however, all methods are evaluated under the same protocol, so comparisons remain fair.

**Mathematical Reasoning Benchmarks**    We evaluate mathematical reasoning on MATH500, AIME24, AIME25, and AMC23 using exact-match accuracy. For each model output, we extract a single final answer from the completion (e.g., the last explicitly stated answer) and apply a normalization step (e.g., stripping formatting tokens and whitespace) before comparison. We count a problem as correct if the normalized extracted answer matches the ground-truth answer provided by the benchmark, and we report accuracy over the full evaluation set. For sampling-based decoding, we apply the same extraction and matching procedure to the selected candidate response.

**MMHal-Bench**    We evaluate multimodal hallucination on MMHal-Bench (Sun et al., 2024b) following the benchmark's official protocol and report the overall score and hallucination rate. Concretely, each response is judged for consistency with the visual evidence, and the hallucination rate measures the fraction of prompts for which the model makes ungrounded claims under the benchmark criteria. In this experiment, we use Qwen3-VL-8B-Instruct as the base model; with preference injection, we place the preference text in the system prompt, and without preference injection, we provide only the image and the corresponding question in the user request.

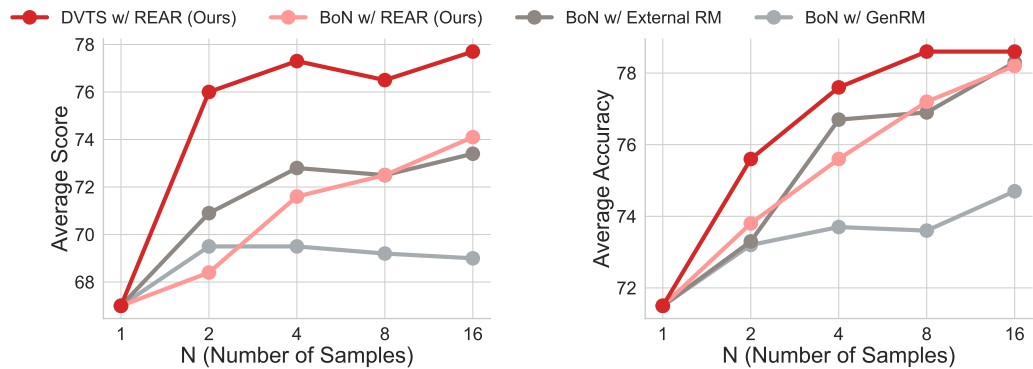

*Figure 6.* Scaling performance on the PrefEval benchmark with varying numbers of samples ($N$) for different methods. We use the average LLM-evaluated scores for the explicit preference task (left) and the accuracy of selected choices for the implicit choice task (right).

*Table 8.* Sensitivity of REAR to the preference text $x_p$ on MATH500 with Qwen2.5-7B-Instruct (BoN, $N = 16$).

| Preference text $x_p$ | Accuracy |
|---|---|
| Greedy decoding (no $x_p$) | 74.6 |
| *Detailed* (used in main experiments): "Act as a rigorous mathematician... double-check your result by working backwards or using a different method." | **82.6** |
| *Brief:* "Solve this math problem step by step." | 81.8 |
| *Unrelated:* "You are a creative fiction writer..." | 74.8 |

## E. Detailed Comparison with Decoding-Time Methods

The token-level form of REAR's reward (Lemma 3.2) is a log-ratio combination of policy probabilities with and without the preference text, which superficially resembles the contrastive log-ratios used in DExperts (Liu et al., 2021), Contrastive Decoding (CD) (Li et al., 2023b), and Drift (Kim et al., 2025). Table 7 summarizes the substantive differences along three axes.

The most consequential difference is the *Role* row. DExperts, CD, and Drift each define a per-token sampling distribution that replaces the base model's logits during autoregressive generation, producing a single output stream. REAR instead defines a scalar score over complete or partial trajectories, which is consumed downstream by a TTS algorithm (best-of-$N$ or tree search) that compares and selects among candidate trajectories. This is why REAR-guided BoN can rank $N$ independently generated responses and why DVTS can use REAR to guide tree expansion at segment boundaries (Algorithms 1 and 2); none of the three decoding methods support either workflow without re-design.

The structural similarity at the token level should also be qualified. The reduction of REAR's reward to a simple log-ratio holds only at the trajectory level, where the potential-based shaping terms $Z(s) - \gamma Z(s')$ in Lemma 3.2 telescope; at the token level the value-function terms remain. Moreover, REAR's $\lambda$ is not a free decoding-temperature parameter: it has the interpretation of the ratio between the test-time and training-time preference strengths in the MaxEnt RL derivation (Proposition 3.1). Drift is the closest related work in spirit but still requires a hand-designed attribute set and preference pairs and operates as a decoding-time logit modifier, whereas REAR is zero-shot, single-text, and trajectory-level.

## F. The Scaling Performance of REAR-guided TTS Methods

We investigate how the performance of our method scales with the number of samples ($N$). As shown in Figure 6, performance on the PrefEval explicit preference and implicit choice datasets improves as $N$ increases, with diminishing returns for larger values. Our BoN approach with REAR demonstrates scaling performance comparable to the variant using an external RM. The DVTS variant achieves stronger performance with a smaller sampling budget, highlighting the efficiency of its step-by-step tree search approach.

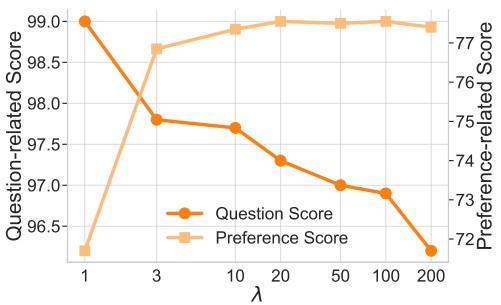

*Figure 7.* Scores on questions and preference of REAR-guided TTS methods on PrefEval explicit preference data with different $\lambda$ values.

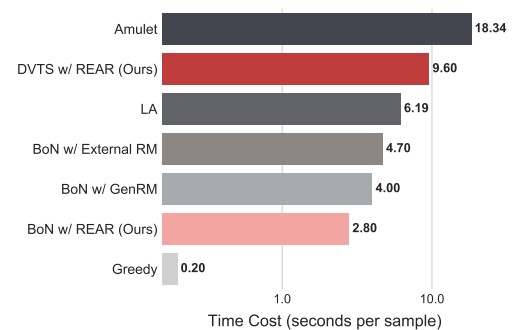

*Figure 8.* Time cost of different methods on the PrefEval explicit preference task.

## G. Sensitivity to the Preference Text

REAR conditions on a free-form preference text $x_p$ rather than a learned attribute set or trained reward model, which raises the question of how the choice of $x_p$ affects performance. We study this on MATH500 with BoN ($N = 16$) on Qwen2.5-7B-Instruct, comparing three preference texts of varying task relevance against greedy decoding (Table 8): a *detailed* version that elaborates rigor and double-checking (the variant used in our main math experiments), a *brief* version that names only the task, and an *unrelated* version that asks the model to write creative fiction.

Two observations follow. First, REAR is robust to rephrasing of a task-relevant preference: the brief version (81.8) drops only 0.8 points from the detailed original (82.6), well above greedy decoding. This indicates that practical use of REAR does not require carefully engineered preference text. Second, REAR does not blindly boost scores from arbitrary text: the unrelated preference yields 74.8, statistically indistinguishable from greedy decoding (74.6). The score function rewards trajectories that match the preference, so a preference unrelated to mathematical reasoning produces no realignment signal in either direction. Together these results suggest that the gains in our main experiments come from REAR exploiting genuine alignment between $x_p$ and the task rather than from $x_p$ acting as a generic generation prior.

## H. Additional Experiments on Hyper-parameter Tuning

### H.1. Trade-off Between Question and Preference Components

To understand this non-monotonic relationship on $\lambda$, we analyze how $\lambda$ affects different aspects of response quality. The detailed rubrics from the PrefEval explicit preference task allow us to disentangle performance into two components: general response quality and preference alignment. We use the "helpfulness" score to measure the former and the average of "preference violation" and "preference acknowledgement" scores for the latter. As illustrated in Figure 7, these two components exhibit monotonic trends with respect to $\lambda$. As $\lambda$ increases, the preference-related score improves, while the question-related score (helpfulness) declines. This trade-off explains why simply increasing $\lambda$ does not guarantee better overall performance; an excessively large $\lambda$ compromises the model's fundamental ability to provide helpful answers, thereby reducing the overall quality of the response.

### H.2. Sensitivity to $\lambda$ on Preference Benchmarks

We conduct an ablation study on the hyper-parameter $\lambda$ in REAR, which controls the weight of the value function. The results are shown in Table 5a and Table 5b. The results largely confirm the observations made in Section 5.4. Across most tasks on the PrefEval and Multifaceted benchmarks, we observe a non-monotonic relationship between $\lambda$ and performance. For the majority of these tasks, the optimal performance is achieved when $\lambda$ is around 20 for both BoN and DVTS. This reinforces the idea that there is a trade-off between adhering to user preference and maintaining the general quality of the response, as an excessively high $\lambda$ can degrade helpfulness.

However, we also note some task-specific variations. For instance, on the PrefEval Explicit Preference task, DVTS achieves its best performance with a smaller $\lambda$ of 3. On the PrefEval Implicit Preference and the Ping-Pong Bench tasks, BoN with REAR shows a trend of continuously improving performance as $\lambda$ increases up to 50. This suggests that for certain tasks, particularly those requiring strong adherence to a persona (Ping-Pong) or subtle preference cues, a stronger emphasis on the

*Table 9.* End-to-end latency and throughput on the PrefEval explicit preference task. Slowdown is relative to greedy decoding.

| Method | Latency (s/sample) | Slowdown | Throughput (samples/s) |
|---|---|---|---|
| Greedy | 0.20 | 1.0× | 5.00 |
| BoN w/ REAR (Ours) | 2.80 | 14.0× | 0.36 |
| BoN w/ GenRM | 4.00 | 20.0× | 0.25 |
| BoN w/ External RM | 4.70 | 23.5× | 0.21 |
| LA | 6.19 | 31.0× | 0.16 |
| DVTS w/ REAR (Ours) | 9.60 | 48.0× | 0.10 |
| Amulet | 18.34 | 92.0× | 0.05 |

preference-related reward component is beneficial.

Furthermore, the optimal range for $\lambda$ appears to depend on the specific TTS algorithm. For example, the DVTS algorithm adopts a step-by-step tree search strategy, which can be more sensitive to the preference reward. Exaggerating the preference reward may lead to suboptimal performance. In contrast, BoN methods only rate the final response after finishing generation, where a large $\lambda$ value is often preferred for the benchmark. For the Ping-Pong benchmark, DVTS achieves its peak performance at $\lambda = 1.0$, while BoN performs best with a much larger $\lambda$. This highlights that the interaction between the search strategy and the reward scaling is an important factor. In summary, while a $\lambda$ of 20 serves as a robust default for many scenarios, fine-tuning this hyper-parameter for the specific task and TTS method can unlock further performance gains.

### H.3. Validation-based $\lambda$ Selection

To rule out the possibility that the $\lambda$ values used in Tables 1 and 2 were selected on the test set, we conduct a validation-based selection experiment on Qwen2.5-7B-Instruct. For each PrefEval and Multifaceted task we split the data 80/20 into test and validation, sweep $\lambda \in \{3, 10, 20, 50\}$ on the validation split, and report the test-split score under the validation-selected $\lambda$. Table 6 compares this against the $\lambda$ used in our main tables. In 7 of 8 configurations the validation procedure re-selects the same $\lambda$; the remaining case (BoN, PrefEval Implicit Preference) prefers $\lambda = 20$ on validation versus $\lambda = 50$ in the main tables, with a 0.2-point test-score difference. This confirms that $\lambda = 20$ is a robust default and that REAR's gains are not the result of test-set tuning.

### H.4. Sensitivity to $\lambda$ on Mathematical Tasks

For the mathematical benchmarks we use $\lambda = 20$ uniformly across tasks and models, without per-task tuning. Table 5c reports the full $\lambda$ sweep for BoN with REAR on Qwen2.5-7B-Instruct: performance varies by less than 1 point across $\lambda \in \{3, 10, 20, 50\}$ on MATH500, AIME25, and AMC23, confirming that the math results are insensitive to $\lambda$ in the studied range.

## I. Time Cost of REAR-guided Test-time Methods

REAR offers significant efficiency gains over baselines that rely on external reward models. We report the inference cost of REAR-guided methods compared to other baselines on the PrefEval explicit preference task in Figure 8 and provide the corresponding numbers in Table 9. Using a node of 8 NVIDIA GPUs with 96GB memory, by calculating rewards from the model's internal probabilities, REAR avoids the substantial computational overhead of loading and executing additional models. This makes REAR-guided methods not only more efficient but also easier for deployment.

Among the test-time methods reported in Table 9, BoN with REAR is the fastest: it is 1.4× faster than BoN with GenRM and 1.7× faster than BoN with an external reward model, since REAR adds only a single non-autoregressive forward pass per response and reuses the same model for generation and scoring (no separate reward model needs to be loaded or queried). DVTS with REAR is more expensive due to its tree-search structure but remains 1.9× faster than the test-time alignment baseline Amulet, which iterates 60 optimization steps per sample.

## J. Prompt Templates of the GenRM Baseline

In this section we provide the prompt templates of the GenRM baseline on preference alignment tasks.

*Listing 1.* Generative Verification Prompting Template

```
System: [Preference in the data sample]

User: [Question]

Assistant: [Response]

User: Please act as an impartial judge and evaluate the quality of the
    assistant's response. A preferred response is helpful, harmless, and
    accurately follows instructions. Is this a preferred response? Answer '
    Yes' or 'No' in the format 'Preferred: X'.

Assistant: [Potential chain-of-thought reasoning process] Preferred: [Yes/No]
```

