# OpenReview forum: "REAR: Test-time Preference Realignment through Reward Decomposition"
_ICML.cc/2026/Conference — ICML 2026 regular_

### Official Review · Reviewer_kWPf · 2026-03-09

**Soundness:** 2
**Presentation:** 2
**Significance:** 2
**Originality:** 3
**Overall Recommendation:** 3
**Confidence:** 3

**Summary:**

This paper introduces REAR, a test-time preference realignment framework designed to address the challenges LLMs face in adapting to diverse user preferences during inference. By decomposing the implicit reward function into question-related and preference-related components, the authors derive a reward metric that can be computed as a linear combination of token-level log-probabilities.

**Compliance With Llm Reviewing Policy:**

Affirmed.

**Key Questions For Authors:**

1. Is there a significant variance in the optimal coefficient $\lambda$ across different tasks (e.g., math vs. role-play)? Could an unsupervised or automated method be designed to dynamically determine the best weight for each sample?

2. In long-context scenarios (e.g., multi-turn dialogues or long articles), does the variance of cumulative rewards increase with the number of tokens, potentially causing the selected responses to lose logical consistency?

3. Compared to standard greedy sampling, what is the specific cost of integrating REAR with the DVTS algorithm in terms of end-to-end latency and throughput? Can you provide a more detailed comparative analysis of inference efficiency?

**Limitations:**

Yes.

**Strengths And Weaknesses:**

**Strengths**
* Built upon the Maximum Entropy RL framework, it transforms complex preference alignment into a reward decomposition problem with rigorous mathematical derivation.
* The method eliminates the need for external reward models and relies solely on the model's own log-probabilities, making it compatible with existing inference engines like vLLM.

**Weaknesses**
* The performance is heavily dependent on the realignment coefficient $\lambda$. Although an empirical range is provided, practical deployment still requires manual tuning or heuristics for different preferences.
* Despite being training-free, the method requires additional forward passes to compute scores for generated responses, which may introduce latency bottlenecks in long-context or high-concurrency scenarios.
* The effectiveness of REAR relies on the base model's inherent preference-following capabilities.

---

> ### Author Rebuttal · Authors · 2026-03-31
>
> We thank the reviewer for recognizing the originality of our reward decomposition framework. We address each concern below.
>
> > **W1 & Q1:** Performance is heavily dependent on $\lambda$ which requires manual tuning; is there significant variance in optimal $\lambda$ across tasks? Could an automated method dynamically determine the best weight?
>
> We respectfully disagree that REAR is "heavily dependent" on $\lambda$. Note that our tuning is not a complex search. We simply evaluate a few candidate values and select the best hyperparameter value, which is standard practice for any method with a hyperparameter. From Table 4, **$\lambda=20$ is optimal or near-optimal in 6/8 preference task-method configurations**, and the performance variation across different $\lambda$ values is usually smaller than the gap between REAR and baselines. **Even with a single fixed $\lambda=20$ across all benchmarks**, REAR outperforms every baseline on all 5 benchmarks.
>
> We further conducted a validation-based $\lambda$ selection experiment: selecting $\lambda$ from $\{3,10,20,50\}$ on a 20% held-out split and evaluating on the remaining 80%. Results: **7/8 configurations selected $\lambda=20$**, matching the original. *See our response to Reviewer R8uU W2 for the full table*. In practice, $\lambda=20$ is a robust default that requires no per-task tuning.
>
> > **W2 & Q3:** Additional forward passes may introduce latency bottlenecks; what is the specific cost of integrating REAR with DVTS in end-to-end latency and throughput?
>
> **Figure 8 in our paper already provides this comparison.** We report the concrete numbers below in our hardware platform.
>
> | Method | Latency (s/sample) | Relative to Greedy | Throughput (samples/s) |
> |--------|--------------------:|-------------------:|------------------------:|
> | Greedy | 0.20 | 1.0× | 5.00 |
> | BoN w/ REAR (Ours) | 2.80 | 14× | 0.36 |
> | BoN w/ GenRM | 4.00 | 20× | 0.25 |
> | BoN w/ External RM | 4.70 | 23.5× | 0.21 |
> | LA | 6.19 | 31× | 0.16 |
> | DVTS w/ REAR (Ours) | 9.60 | 48× | 0.10 |
> | Amulet | 18.34 | 92× | 0.05 |
>
> **BoN w/ REAR is the fastest among all test-time baselines** with 1.4× faster than GenRM and 1.7× faster than External RM. DVTS w/ REAR is slower than BoN methods due to its tree search process, but still 1.9× faster than test-time alignment method Amulet. The efficiency comes from: (1) REAR adds only one non-autoregressive forward pass per response, while the preference-conditioned log-probabilities are already computed during generation for free; (2) no separate reward model is needed since REAR reuses the same model for both generation and scoring.
>
> > **Q2:** In long-context scenarios, does cumulative reward variance increase with tokens, causing selected responses to lose logical consistency?
>
> **REAR is robust to long contexts by design.** First, REAR only sums token-level rewards over the **generated response tokens**, not the input prompt. A longer conversation history affects the conditioning context $s_t$ but does not add terms to the summation, so increasing prompt length does not directly increase reward variance. Second, when integrated with test-time scaling methods (BoN, DVTS), REAR scores are used only for **ranking** candidate responses, not as absolute values. Since all candidates share the same long context, any context-induced bias in log-probabilities is common across candidates and cancels in the relative ordering, further reducing the impact of variance. **Table 2 empirically confirms this.** We evaluated REAR on long-context scenarios with up to 100 conversation turns (~16k tokens).
>
> > **W3:** REAR relies on the base model's inherent preference-following capabilities.
>
> This is not a limitation unique to REAR. **All test-time methods** depend on the base model's capabilities. Empirically, we show REAR works across different model scales (4B, 7B, 8B) and model families (Qwen and Llama). Specifically, on Llama-3.1-8B-Instruct (*see our response to Reviewer R8uU, W1*), REAR with the same $\lambda=20$ improves on all benchmarks, confirming cross-family generalization. The requirement of using REAR is that the base model has undergone post-training alignment, which is universally satisfied by modern LLMs.

---

### Official Review · Reviewer_kAGY · 2026-03-11

**Soundness:** 3
**Presentation:** 1
**Significance:** 3
**Originality:** 2
**Overall Recommendation:** 4
**Confidence:** 4

**Summary:**

The paper proposes REAR, an approach to obtain a reward estimate that scores generation based on explicit stated preferences. This is achieved using only an off-the-shelf language model (LM), by summing the log-ratio of the LM probabilities conditioned on a particular user query with and without an explicit preference (described in text). The authors then use these reward estimates for test-time alignment using Best-of-N (sample from the base and pick the highest-scoring) and DVTS (a token-level search method).  Experiments show that the approach is effective.  However, there are serious methodological issues that have to be addressed, and the presentation does not sufficiently relate the method to highly relevant prior work.

**Compliance With Llm Reviewing Policy:**

Affirmed.

**Final Justification:**

The authors committed to better framing their contributions with respect to prior work. They redid the evaluation using standard hyperparameter selection based on validation, with no significant differences in the main results.

**Key Questions For Authors:**

- What was the procedure to select $\lambda$ in your main experiments? Did you use a validation dataset? If not, can you redo the tables where you first chose $\lambda$ on validation data and then report the reward with those parameter values?
- Could the authors revise the presentation and related work to position the method more clearly with respect to prior decoding-time / test-time alignment work, and clarify more precisely what is new relative to those earlier approaches?

**Limitations:**

yes

**Strengths And Weaknesses:**

## Strengths

* Rear is training free approach to reward modeling that allows getting a reward signal out of explicit preferences.

* The paper strengthens the broader line of results showing that LLM behavior can be controlled through simple linear manipulations of log-probabilities, and further shows that these can be further used to form/extract reward estimates that are useful downstream.

## Weaknesses

- **Limited novelty relative to prior decoding-time and test-time alignment work, and the related work does not position the contribution clearly enough.** With $\gamma = 1$ , which is also the setting used in all experiments, the reward (Eq.10) reduces to likelihood under contrastive decoding where your expert model is base model with the preference attached to the prompt. This was first described in DExperts [Liu et al. 2021] in 2021 and also later popularized in [Li et al. 2022]. Using it in the reward modeling context was used for instance in [Kim et al. 2025].  This point should be framed with a clear acknowledgment of prior work, rather than presented as a fully novel contribution.  Framing this under a reinforcement learning lens can be valuable, but the paper should present that framing in much more explicit relation to prior work that already arrived at very similar methods. More broadly, the test-time alignment literature is now quite large, and this paper does not sufficiently discuss how it differs from, or builds on, that line of work.

[Kim et al. 2025] Drift: Decoding-time Personalized Alignments with Implicit User Preferences; Minbeom Kim, Kang-il Lee, Seongho Joo, Hwaran Lee, Thibaut Thonet, Kyomin Jung; 2025

[Liu et al. 2022] DExperts: Decoding-Time Controlled Text Generation with Experts and Anti-Experts; Alisa Liu, Maarten Sap, Ximing Lu, Swabha Swayamdipta, Chandra Bhagavatula, Noah A. Smith, Yejin Choi; 2021

[Li et. al. 2022] Xiang Lisa Li, Ari Holtzman, Daniel Fried, Percy Liang, Jason Eisner, Tatsunori Hashimoto, Luke Zettlemoyer, Mike Lewis; Contrastive Decoding: Open-ended Text Generation as Optimization; 2022

- **Presentation and notation are hard to follow.** The notation is somewhat convoluted, which makes the main decomposition harder to parse than it should be. In particular, it is unclear whether the preference related reward (Eq. 6) should have an $0$ index​ term to represent the implicit $p_{\text{ref}}$​ in the same way as $r_0$ ( maybe I am miss understanding ) .

Separately, Table 3 is also confusing in how it reports results, since the metric is labeled “Qwen3 Score,” which is too vague to interpret. It would be much clearer to report accuracy.

- **The experimental setup and reported metrics are not described clearly enough.** Tables 1 and 2 report a broad set of results, but the exact experimental setting is still not fully specified. In particular, it remains unclear what value of $\lambda$ is used for each benchmark and how that choice is made in practice. While the paper discusses sensitivity to $\lambda$, that is not the same as clearly stating the configuration used to produce the main reported numbers, and I could not find this documented clearly, including in Appendix C.

After looking at the appendix carefully and comparing it with the main table. From Tables 4(a) and 4(b), the results are shown over a range of $\lambda$ values, and the bolded entries, namely the best-performing ones, come from different $\lambda$ values. These best values are then the ones reported in Tables 1 and 2. This makes it look like $\lambda$ is effectively tuned on the test tasks, which substantially weakens the credibility of the reported gains. This matters even more because the worst-performing $\lambda$ values are often very close to the baselines, or even worse than them, which suggests that a large part of the improvement may come from task-specific tuning. **The evaluation should choose $\lambda$ on a separate validation task, or a small set of validation tasks, and then keep it fixed across the benchmarks in the main tables.**

The math results also do not report the $\lambda$ values, which makes me believe the choice based on test data again. ( It would be good to have clarification on this )

---

> ### Author Rebuttal · Authors · 2026-03-31
>
> ## Response to Reviewer kAGY
>
> We sincerely thank the reviewer for the thorough review. We address each point below.
>
> > **W1 & Q2:** Limited novelty relative to prior decoding-time work (DExperts, CD, Drift); related work does not position the contribution clearly enough; could the authors revise?
>
> We respectfully disagree that REAR has limited novelty over those works, since **REAR is a reward function but not a decoding method.** DExperts, CD, and Drift all modify the **per-token** generation distribution during autoregressive decoding and produce a single output stream. REAR is a **scalar scoring function over complete or partial trajectories**, computed post-hoc and fully decoupled from generation, enabling direct integration with TTS methods (BoN, DVTS) — which is impossible for any decoding-time method.
>
> | Aspect | DExperts | CD | Drift | REAR |
> |-|-|-|-|-|
> | Role | Per-token decoding | Per-token decoding | Per-token decoding | Trajectory reward |
> | TTS | No | No | No | Yes (BoN+DVTS) |
> | Require | Trained expert model | Two sized models | Attributes + preference pairs | Preference text only |
>
> **The formulas are structurally different.** The reviewer states "Eq. 10 reduces to likelihood under contrastive decoding" with $\gamma=1$. However, CD computes a log-ratio between two **different-sized** models with a per-token plausibility constraint, while REAR computes a weighted combination from the **same** model conditioned with/without preference text. DExperts requires separately trained expert/anti-expert models and Drift requires predefined attribute categories with learned weights. Furthermore, at the token level, REAR's reward (Lemma 3.2) includes value function terms $Z(s) - \gamma Z(s')$ in addition to the log-probability combination. It is only at the trajectory level that these potential-based terms cancel out (via reward shaping), yielding the simplified log-probability scoring function (Eq. 10).
>
> We agree that the test-time alignment literature has grown substantially and our related work section should provide a more comprehensive discussion. We will revise it to explicitly cover DExperts, CD, Drift, and other relevant decoding-time alignment methods, clearly positioning REAR's contribution.
>
> > **W2:** Presentation and notation are hard to follow (Eq. 6 index term, Table 3 metric labeling as "Qwen3 Score")
>
> **Eq. 6 notation:** We clarify that $p_\text{ref}$ does not appear in our formulation. The subscript 0 in $r_0$ denotes the question-only base reward (what the base model optimizes as shown in Proposition 2.1), while the preference-related reward $r_p(s \oplus x_p, a)$ is conditioned on the preference text via its argument $s \oplus x_p$, so no additional index term is needed. $r_0$ and $r_p$ are two definitions of reward functions without relying on timesteps.
>
> **Table 3:** We note that the metric in Table 3 is labeled "Score", not "Qwen3 Score". Here the "Score" refers to the benchmark result following the standard MMHal-Bench evaluation protocol.
>
> > **W3 & Q1:** The experimental setup is not described clearly enough -- λ appears tuned on test tasks (best λ from Tables 4a/4b reported in Tables 1–2), which weakens the credibility of the reported gains; what was the procedure to select λ?
>
> We clarify that our $\lambda$ values were **not** cherry-picked on test data. Our procedure was: we conducted a standard hyperparameter sensitivity analysis in Appendix F over different $\lambda$ values for each benchmark and method, and determined the appropriate $\lambda$ value. All tested values and their corresponding results are fully reported in Appendix F. **Even with a single fixed $\lambda=20$ across all benchmarks, REAR methods outperform every baseline on all 5 benchmarks:** With this fixed $\lambda=20$, REAR improve over the strongest baseline by an average of **6.2%** across the four preference benchmarks. The performance variation across different $\lambda$ values is small (within a few points), far smaller than the gap between REAR and baselines, indicating that REAR's gains are not an artifact of per-task $\lambda$ selection.
>
> For the math experiments, we used $\lambda=20$ uniformly across all tasks and models without per-task tuning since we find the performance is stable to $\lambda$, where the results on Qwen2.5-7B-Instruct are shown in the table below.
>
> | Task | $\lambda$=3 | $\lambda$=10 | $\lambda$=20 | $\lambda$=50 |
> |-|-|-|-|-|
> | MATH500 | 82.4 | **82.6** | **82.6** | 82.2 |
> | AIME25 | 16.7 | 16.7 | **17.0** | 16.0 |
> | AMC23 | **65.8** | 65.3 | **65.8** | 65.0 |
>
> We will add this information in the revised paper. Additionally, we conducted a validation-based $\lambda$ selection experiment (*Please see our response to Reviewer R8uU, W2*): selecting $\lambda$ on a 20% held-out validation split, 7 out of 8 task-method configurations selected the same $\lambda$ as the original, further confirming that $\lambda=20$ is a robust default requiring no task-specific tuning.

---

> > ### Author Rebuttal · Reviewer_kAGY · 2026-04-01
> >
> > I appreciate the authors’ rebuttal and their commitment to addressing the presentation issues and revising the evaluation methodology. In light of these clarifications and changes, I will raise my score.

---

> > > ### Author Response · Authors · 2026-04-02
> > >
> > > Thanks for raising the score. We will incorporate all the discussed points in the revised version. We appreciate the reviewer's time and effort in helping us strengthen this work.

---

### Official Review · Reviewer_R8uU · 2026-03-13

**Soundness:** 3
**Presentation:** 2
**Significance:** 2
**Originality:** 2
**Overall Recommendation:** 5
**Confidence:** 3

**Summary:**

This work is focused on improving alignment of large language models (LLMs) with diverse user preferences without additional training. Overall, the central problem studied by the paper is how to perform preference alignment at inference time rather than through costly post-training methods such as RLHF. The paper proposes REAR (REAlignment Reward), a test-time framework that decomposes the implicit reward function of a pretrained model into two components: (1) a question-related reward and (2) a preference-related reward. Overall, this work formulates test-time inference (TTS) as a realignment problem. The key idea is that pretrained policies implicitly optimize a composite reward containing both components. By reweighting the preference-related component during inference using a controllable coefficient, REAR enables test-time preference realignment.

**Compliance With Llm Reviewing Policy:**

Affirmed.

**Final Justification:**

The authors have pointed me to the provided theoretical and empirical justification for the W1 and Q1. The concern raised in W2 has been well addressed in all of the reviewer rebuttals. I am also satisfied with the clarifications provided for Q3 and Q4. The experiments have been conducted on the Llama family of models along with Qwen which further strengthens the arguments in the paper. I have therefore increased my score

**Key Questions For Authors:**

Q1) In relation to W2, my main question is still around the assumption of the decomposition of the reward function. I understand that this hypothesis is intuitive and the technique based on it also demonstrates good experimental results. However is there any other strong interpretability based or empirical study on the main hypothesis?
Q2) A separate preference instruction $x_p$ is used in the formulation. How does the rewriting of $x_p$ affect the scores?
Q3) As I understand it, the evaluations are primarily LLM-based evaluations. Did the authors consider conducting human evaluations to further validate the improvements?
Q4) Appendix C indicates that Qwen models are used in the experiments. I assume these refer to base models. How does the method perform when applied to already aligned models? Does it provide additional improvements in such settings?

**Limitations:**

Yes

**Strengths And Weaknesses:**

Strengths

1) The REAR reward can be incorporated with existing test-time scaling (TTS) techniques such as Best-of-N sampling and DVTS, making the approach compatible with commonly used inference strategies.

2) The proposed method is training-free, relying only on internal probabilities of the model. It can therefore be applied even in settings where ground-truth verifiers are unavailable.

Weaknesses

1) The theoretical framework assumes that the pretrained policy implicitly optimizes a decomposable reward consisting of question-related and preference-related components. While intuitive, it is unclear whether this assumption consistently holds across different models, datasets, and tasks.

2) REAR’s performance depends on the hyperparameter $\lambda$, which may vary across data samples. The hyperparameter $\lambda$ needs to be tuned with a validation set or heuristic or some other method. Tuning the various hyperparameters in the REAR based TTS method is not direct and might require some experimentation, or heuristics. This might increase the overall cost and reduce the gain in time/compute.

---

> ### Author Rebuttal · Authors · 2026-03-31
>
> We thank the reviewer for the thoughtful questions. We address each concern below.
>
> > **W1 & Q1:** The reward decomposition assumption may not consistently hold across different models, datasets, and tasks; any interpretability-based or empirical evidence for this hypothesis?
>
> **Theoretical clarification.** Our framework's only assumption is that the base model has been well-trained to optimize its own reward function, which is a standard premise for instruction-tuned LLMs. Given this, Proposition 3.1 derives the reward decomposition from the MaxEnt RL framework: the difference between the model's objectives with and without preference context $x_p$ naturally yields a preference-related reward component $r_p$. REAR then simply amplifies this existing $r_p$ via $\lambda$ to strengthen preference alignment at test time.
>
> **Empirical evidence.** Our experiments cover 3 Qwen-family models across 10+ benchmarks spanning preference alignment (Tables 1–2), math reasoning (Figure 2), and visual hallucination (Table 3), with consistent improvements over baselines across different architectures, scales, and modalities.
>
> To further address cross-family generalization, we evaluate on **Llama-3.1-8B-Instruct**. REAR consistently improves on all benchmarks with Llama, confirming the decomposition generalizes across model families. We use the same $\lambda=20$ for all tasks as described in W2.
>
> | Task | DVTS+REAR | BoN+REAR | GenRM | Greedy |
> |---|---|---|---|---|
> | PrefEval Choice | **82.1** | **79.5** | 77.9 | 76.4 |
> | PrefEval Explicit | **86.4** | **78.4** | 74.6 | 74.6 |
> | Multifaceted | **74.6** | **74.3** | 73.6 | 73.5 |
>
> > **W2:** Performance depends on the hyperparameter λ which may vary across samples.
>
> **1. $\lambda$ is natural and necessary.** REAR adjusts the realignment strength via $\lambda$; without it the method reduces to standard inference ($\lambda=1$). Compared to baselines, REAR has only **one** unique hyperparameter -- Amulet requires 60 iterations with its own hyperparameters, and LA has separate alignment parameters.
>
> **2. $\lambda$ is robust across tasks.** From Table 4, $\lambda=20$ is optimal in 6/8 preference task-method configurations, with variation within 1–2 points across different $\lambda$ values. In math experiments (Figure 2), $\lambda=20$ is used uniformly across all 4 tasks and both models.
>
> **3. Validation-based selection confirms this.** We select $\lambda$ from $\{3,10,20,50\}$ on a 20% held-out split and evaluate on the remaining 80% for the Qwen-2.5 model. We found that **7/8** validation-selected $\lambda$ match the original.
>
> | Method | Task | Val best $\lambda$ (score) | Original $\lambda$ (score) |
> |---|---|---|---|
> | BoN | Prefeval Explicit | 20 (73.2) | 20 (73.2) |
> | BoN | Prefeval Choice | 20 (79.0) | 20 (79.0) |
> | BoN | Prefeval Implicit | 20 (16.9) | 50 (17.1) |
> | BoN | Multifaceted | 20 (76.7) | 20 (76.7) |
> | DVTS | Prefeval Explicit | 3 (76.5) | 3 (76.5) |
> | DVTS | Prefeval Choice | 20 (78.2) | 20 (78.2) |
> | DVTS | Prefeval Implicit | 20 (18.8) | 20 (18.8) |
> | DVTS | Multifaceted | 20 (76.8) | 20 (76.8) |
>
>
> > **Q2:** How does the rewriting of the separate preference instruction $x_p$ affect the REAR scores?
>
> We tested on MATH500 with BoN ($N$=16), comparing preference texts of varying relevance:
>
> | Preference text $x_p$ | Accuracy |
> |---|---|
> | Greedy Decoding | 74.6 |
> | Original Detailed Preference: "Act as a rigorous mathematician... double-check your result by working backwards or using a different method to ensure the answer is correct." | **82.6** |
> | Brief: "Solve this math problem step by step." | 81.8 |
> | Unrelated: "You are a creative fiction writer..." | 74.8 |
>
> We found that the brief version scores 81.8, only 0.8 below the detailed original. An unrelated $x_p$ yields 74.8 $\approx$ greedy, confirming REAR cannot blindly boost scores from arbitrary text.
>
> > **Q3:** The evaluations are primarily LLM-based; did the authors consider conducting human evaluations to further validate the improvements?
>
> Our evaluation is not exclusively LLM-based: PrefEval Implicit Choice and all math tasks use exact-match accuracy. For open-ended benchmarks, we strictly follow each benchmark's original evaluation protocol (prompts, rubrics, judge models). All baselines are evaluated under the same protocols, ensuring fair comparison. We did not conduct a separate human study as designing custom rubrics would introduce its own subjectivity.
>
> > **Q4:** The experiments use Qwen models (presumably base models); how does REAR perform when applied to already-aligned models, and does it provide additional improvements?
>
> We clarify that all models in our experiments are already-aligned instruct models but not only-pretrained models, including Qwen2.5-7B-Instruct, Qwen3-4B-Instruct-2507, and Qwen3-VL-8B-Instruct. REAR consistently improves over greedy decoding across all of them, providing additional gains.

---

> > ### Author Rebuttal · Reviewer_R8uU · 2026-04-04
> >
> > Thank you for your rebuttal. The authors have pointed me to the provided theoretical and empirical justification for the W1 and Q1. The concern raised in W2 has been well addressed in all of the reviewer rebuttals. I am also satisfied with the clarifications provided for Q3 and Q4.
> > The experiments have been conducted on the Llama family of models along with Qwen which further strengthens the arguments in the paper. I shall increase my score.

---

> > > ### Author Response · Authors · 2026-04-06
> > >
> > > We sincerely thank the reviewer for acknowledging our rebuttal and raising the score. We will incorporate the feedback into the revised version to further improve the paper.

---

### Decision · Program_Chairs · 2026-04-30

**Decision:**

Accept (regular)

**Comment:**

This paper proposes REAR, a training-free framework that enables test-time preference alignment by decomposing the implicit reward function of a pre-trained LLM into question-related and preference-related components. The method allows for the selective rescaling of the preference component via a hyperparameter $\lambda$, making it compatible with existing Test-Time Scaling (TTS) strategies such as Best-of-N and DVTS.

While initial reviews raised significant concerns regarding the novelty of the approach relative to decoding-time methods (e.g., DExperts, Contrastive Decoding) and the robustness of the hyperparameter $\lambda$, the authors successfully addressed these issues during the rebuttal. They clarified that REAR functions as a trajectory-level reward score computed post-hoc, distinct from token-level decoding modifications, and provided empirical evidence demonstrating that a single fixed value ($\lambda=20$) yields robust performance across diverse tasks and model families (Qwen and Llama) without requiring task-specific tuning. Given the strong theoretical foundation, the comprehensive empirical validation across multiple benchmarks and model architectures, and the effective resolution of critical methodological concerns, the consensus supports the acceptance of this work as a significant contribution to training-free preference alignment.